**communications** sustainability

# Riparian vegetation reduces coastal turbidity

Hilary D. Brumberg [1,2,3] ✉, Laura E. Dee [4], Hikari Murayama [5,6], Juan José Alvarado Barrientos [7,8], Brooke Bessesen [9], Marie G. Bouffard [10], Matthew G. Burgess [11], Jorge Cortés [7,8], Samuel Furey [12], Noelia Hernández [13], Alexa M. Luger[1], Marguerite Madden [14], Emily Pauline[15], Rafael J. P. Schmitt [16], Katherine J. Siegel [17,18,19], Lucía Vargas-Araya[7], Andrew Whitworth [13,20] & Peter Newton [1]

Marine ecosystems worldwide are increasingly degraded by upstream land use activities, compounding climate change impacts. However, empirically quantifying causal land-sea linkages remains challenging. Using remote sensing data (1987-2019) and four causal inference methods, here we developed an empirical and scalable framework to estimate how land use affects coastal turbidity across spatial scales in southern Costa Rica. We found that riparian natural vegetation (15 m buffer) significantly reduced gulf turbidity up to 800 m offshore, which overlaps with coral reefs and seagrass habitats. In contrast, pasture and gravel roads increased coastal turbidity. Effects were greatest for rivers that are short, steep, or have low discharge. Watershed-scale land uses showed no significant effects. We provide a replicable, scalable framework to identify causal pathways from land to sea, particularly valuable in data-limited regions. Riparian conservation and restoration could serve as effective strategies to align human land use needs with terrestrial, freshwater, and marine conservation.

Land-derived stressors degrade marine ecosystems globally and compound the impacts of climate change[1-4]. In particular, rivers serve as the primary means of transport of sediments, nutrients, and pollutants from land to the ocean[5-10]. Sediment production from anthropogenic disturbances has risen nearly fivefold since 1950[9] and sediment export from non-vegetated watersheds can exceed forested catchments by up to 200-fold[11]. Elevated sediment and nutrient loads reduce water clarity, smother benthic organisms, promote algal overgrowth, and diminish reef resilience[12-14]. Sedimentation affects up to 41% of coral reefs globally[15] and negatively impacts reef fish populations, seagrass beds, marine food webs, and ultimately threatens coastal protection and livelihoods[14,16,17]. Because turbidity is a shared exposure pathway across coral reefs, seagrass meadows, and other nearshore habitats, understanding how land conservation or degradation influences turbidity has multi-ecosystem relevance[10,18-22]. Yet the pathways through which land use propagates downstream and the spatial scales at which these effects occur remain incompletely understood, particularly in tropical regions where empirical data are scarce and long-term monitoring is limited[23-25].

Historically, terrestrial and marine systems have often been managed separately, despite the ecological and biophysical processes that connect them[17,26]. Land-sea interaction research seeks to understand these linkages, while ridge-to-reef conservation applies this knowledge to integrated management and policy. Ridge-to-reef frameworks have emerged as effective approaches for coordinating terrestrial and marine conservation and can yield more cost-effective and ecologically impactful outcomes than single-realm strategies[17,20,26-29]. Evidence-based prioritization of ridge-to-reef management strategies are important given limited resources and the need to balance human land and resource use with terrestrial, freshwater, and marine conservation objectives[17,26]. Modeling studies suggest sediment delivery to coral reefs is sensitive to landscape spatial configuration[15], and that forest restoration benefits for marine ecosystems are can vary by several orders of magnitude[1,30]. Most previous research has focused on well-monitored systems such as the Great Barrier Reef and temperate ecosystems, with far fewer studies in other tropical regions where coral reefs and other vulnerable ecosystems occur[5,6,23-26,29,31-37].

Given the importance of spatially prioritizing land conservation to meet marine conservation objectives—and the role of rivers as key connectors between land and sea—rivers and riparian buffers represent a potentially underexplored strategy in land-sea interaction research and management. Riparian zones, the interface between land and rivers, intercept sediments, nutrients, and pollutants before they enter waterways[26,38]. Natural vegetation in riparian zones, known as riparian vegetation buffers,

---

have been found to be effective strategies to balance human land use needs with terrestrial and freshwater conservation objectives. They stabilize streambanks, moderate water temperatures, and buffer disturbances such as livestock grazing, roads, and agricultural erosion, decreasing sediment loads into streams by up to 90%[39–42]. Several studies have found that riparian land use has a larger effect on river water quality than land use across entire watersheds[43–46]. Nevertheless, most land-sea interaction research evaluates land use only at watershed scales, and therefore seldom distinguishes riparian from broader catchment effects[3,24,25,47]. Studies connecting river discharge to marine ecosystems typically do not trace these conditions back to specific upstream land use drivers[13,48–50]. As a result, the link between where sediment originates within a watershed (ex. riparian areas) and how far its effects propagate throughout coastal-marine waters remains weakly resolved in many land-sea assessments, particularly in regions where coupled catchment-marine models cannot be fully parameterized or validated[13,48–52].

Several other challenges constrain the land-sea interaction literature and thus the ability to quantify land use impacts on marine ecosystems and inform ridge-to-reef management. First, isolating the causal effects of specific human activities and land uses on marine ecosystems remains challenging due to the presence of confounding factors in the landscape, such as the patchwork of other land uses, roads, and physical characteristics like slope and soil[26,53–55]. Most land-sea interaction studies rely on correlational rather than causal analyses, limiting their ability to inform targeted management actions[13,20,53,56,57]. Other fields have developed causal inference methods that enable robust evaluations of causal relationships in observational data, but these statistical techniques are seldom applied in ecology, especially not in the literatures on land-sea interactions or freshwater ecology[58,59]. Catchment and hydrodynamic models have advanced the understanding of sediment export from catchments to the ocean, and have in some cases been used to study the impact of drivers (e.g. land use change) on sediment export[10,23,34,55,57,60–62]. Yet, such models often require extensive field, hydrological, and oceanographic data and are difficult to parameterize or empirically validate in data-limited systems[10,23,34,57,60,61]. This hinders the ability to clearly isolate the causal effect of specific land use changes or interventions, especially in remote, data-limited regions with sparse

monitoring. Moreover, few modelling efforts include long-term empirical validation of upstream drivers and downstream responses, and parameter uncertainty often remains high[10,23,34,57,60,61]. Ground-based field studies, meanwhile, are typically restricted in spatial extent and duration. Consequently, most land-sea studies remain constrained to a limited number of case-study watersheds, rely heavily on models that often carrying high uncertainties, and rarely include multi-decadal datasets to track change over time[5,6,23,24,26,29,31–35]. Due to these limitations and others, empirical land-sea interaction research is often infeasible in remote tropical regions[24,25,36]. Third, knowledge gaps persist regarding the spatial scales over which terrestrial management actions influence downstream marine ecosystems—such as the offshore extent of riverine conservation impacts—which constrains the design of integrated multi-ecosystem conservation strategies[63]. Even in the Great Barrier Reef, one of the world's most intensively studied land-sea systems, a recent synthesis (Wilkinson et al., 2022) noted that although the key biophysical drivers of sediment and nutrient export are qualitatively understood, few studies have quantified their relative effects, examined long-term changes, or established causal links between land-use change and marine water quality.

To address these gaps, we apply a scalable empirical approach to quantify and disentangle causal impacts of land use at multiple spatial scales suitable for data-limited contexts, by integrating causal inference modelling and remote sensing. Our approach provides a complementary, observation-based method that can be applied where detailed process-based catchment or hydrodynamic models are infeasible, where long-term empirical datasets are unavailable, or where confounding limits their ability to isolate the effects of land use change. The objectives of this paper are to: 1) disentangle the relative influence of riparian versus watershed land use on coastal water quality, 2) apply causal inference methods to isolate land use effects while controlling for confounding physical and land use variables, and 3) demonstrate a scalable empirical framework for quantifying land-sea interactions in data-limited regions. We use the Golfo Dulce, a tropical fjord-like gulf situated in Costa Rica (Fig. 1), as a case study due to its high marine biodiversity and changing land uses over recent decades[64–67]. We ask: How did land use affect turbidity in Golfo Dulce at different spatial scales?

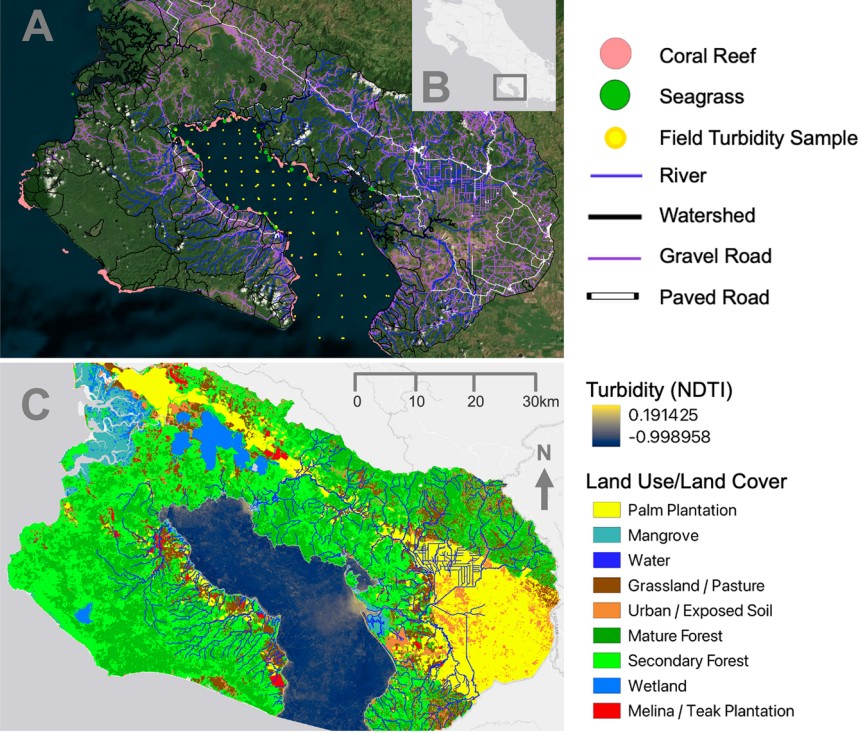

**Fig. 1 | Study area map and visualization of key variables. A** Study area and focal variables. Coral reefs (coral circles)[68,95,97,98], seagrass beds (green circles)[95,96], and field turbidity measurements (yellow). All rivers that flow into the Golfo Dulce (dark blue lines), watersheds (black lines), gravel roads (purple lines), paved roads (white lines). **B** Location of the study area in panel A within Costa Rica. **C** Land use/land cover and turbidity (Normalized Difference Turbidity Index: NDTI) from 2019, along with all rivers that flow into the Golfo Dulce (dark blue). Yellow indicates higher turbidity, and darker blue indicates lower turbidity. A zoomed-up inset map of the Rincón-Riyito River mouth and an aerial photograph are available in Supplementary Fig. 1 to help illustrate the relationship between land use, turbidity plumes, coral reefs, and seagrass beds. Basemaps are from ESRI (Intellectual property of Esri and is used herein under license. Copyright © 2025 Esri and its licensors. All rights reserved.).

**Fig. 2 | Estimated effects of riparian and watershed land use on coastal turbidity.** Estimates from inverse probability of treatment weighting (IPW) models with random effects of the impact of the proportion of A natural vegetation, **B** pasture, **C** plantation, **D** exposed land, and the presence of **E** gravel road, and **F** paved road in riparian zones and watersheds on annual turbidity 100 m offshore. Bars indicate 95% confidence intervals (CI). Blue indicates a p-value of the estimate less than 0.05, and gray indicates a p-value greater than 0.05. Estimates are in Supplementary Table S1.

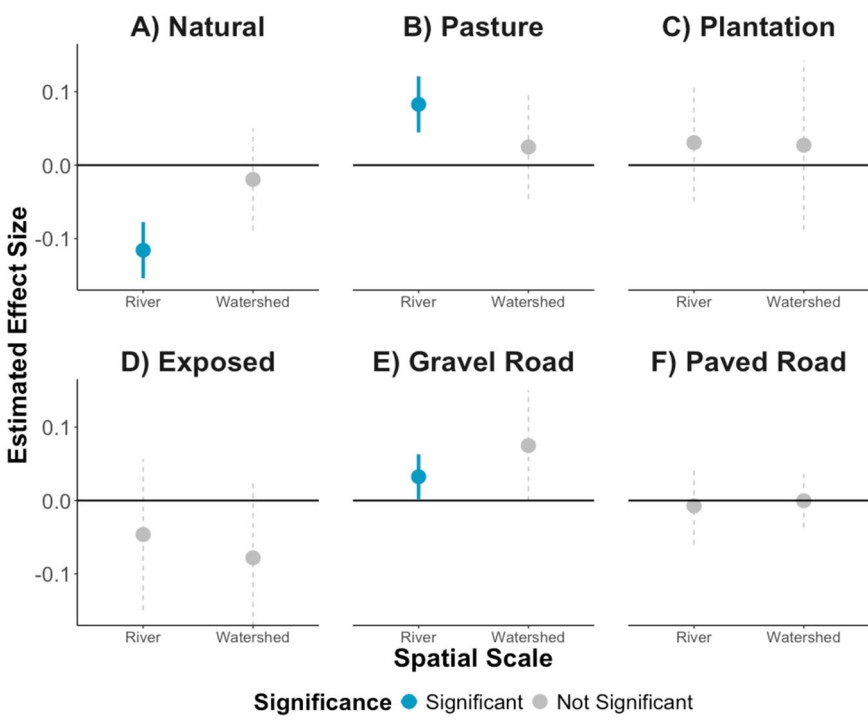

We used land use and land cover (LULC) and Gulf turbidity data (Normalized Difference Turbidity Index, NDTI) derived from open-source remotely sensed imagery from 1987–2019, both of which we validated. These were complemented by additional variables potentially influencing turbidity, including gravel and paved roads. We quantified the effects of land use at two terrestrial spatial scales—entire watersheds and 15-meter-wide riparian zones—on gulf turbidity across 14 offshore distances (25–800 m). To estimate causal impacts, we applied four complementary causal inference models—inverse probability of treatment weighting (IPTW), two-way fixed effects, group mean covariate, and group mean centered—alongside naïve and mixed-effects models, as a robustness check because each model makes different assumptions. IPTW was our primary approach because it best balanced covariates across treatment groups while retaining a flexible, interpretable framework suited for our spatially heterogeneous, observational dataset. Our analysis quantifies causal relationships between terrestrial land use and coastal turbidity. From this, we then interpret potential implications for marine ecosystems such as coral reefs and seagrasses.

We have four main findings: 1) natural vegetation decreased turbidity, while pasture and gravel roads increased it; 2) turbidity impacts were significant only at the riparian scale (15 m buffer), not at the watershed scale, especially in steeper, shorter, low-discharge rivers; 3) land-use impacts extended at least 800 m offshore, affecting coral reefs and seagrass habitats[66–68]; and 4) increased natural vegetation and reduced turbidity align with forest conservation policies. We compare seven modelling approaches to illustrate robustness. These findings help inform ridge-to-reef management by identifying riparian zones as priority areas for balancing terrestrial, freshwater, and marine conservation goals alongside competing land uses. More broadly, our study advances land-sea interaction research by providing a replicable empirical framework that reveals where and how land use most strongly affects marine water quality. To do so, we integrate approaches from causal inference and remote sensing to strengthen the evidence base for spatially-explicit conservation planning.

## Results & discussion
### Natural vegetation decreased gulf turbidity, while pasture and gravel roads increased turbidity

Causal inference analysis helps us to disentangle and quantify the impacts of each land use. We found that increased riparian natural vegetation cover (e.g., forests, wetlands) decreased average annual turbidity (Normalized Difference Turbidity Index, NDTI) 100 m offshore by −0.116 (95% confidence interval: −0.154 to −0.078) (Fig. 2A; Supplementary Table S1). This corresponds to −1.15 standard deviations from the mean, a large effect. Thus, increasing riparian natural vegetation by 10% would decrease the NDTI index by 0.0116, which is −0.115 standard deviations from the mean. This finding suggests that conserving and restoring natural vegetation directly along rivers may help protect coastal ecosystems that are sensitive to turbidity. Because this result was derived from causal inference models using inverse probability of treatment weighting (IPW), it indicates that the observed benefit is attributable to the presence of natural vegetation itself, rather than merely the absence of more degradative land use practices. This result is consistent with previous studies that have found a negative relationship between natural vegetation and turbidity and a positive relationship between natural vegetation and coral reef condition[26,45,69–71].

Several mechanisms may explain how natural vegetation reduces coastal turbidity. Roots stabilize riverbanks and prevent erosion[71]; riparian natural vegetation promotes complex and biodiverse hyporheic zones that enchance sediment filtration[72]; and fallen tree trunks, branches, and leaves create natural dams that retain sediment[26]. Mangroves, which were included in the natural vegetation class, showed a similar negative effect on turbidity but with a wider confidence interval (Supplementary Fig. 2). This greater uncertainty likely reflects the small extent and environmental variability of mangrove habitats, where tidal dynamics and geomorphology may vary substantially among sites. Nonetheless, the direction of the relationship supports the sediments-retention of function of mangroves[26,73].

In contrast, increases in riparian pasture cover increased NDTI (estimate: 0.083, 95% CI: 0.044 to 0.121) and may exert a more consistent effect on turbidity than other human land uses such as plantations and exposed land (Fig. 2). This impact may be due to several factors; pasture grasses typically have shallow root systems that are less effective at stabilizing soil

and reducing erosion, and they often reduce stormwater infiltration due to soil compaction[26]. Livestock can further exacerbate erosion through trampling and soil disturbance, as well as by eroding river banks, mobilizing sediments, and leaving waste when entering rivers to cross or drink[39,42]. Similarly, other *in situ* studies across the Neotropics found that the pastures degrade water quality in rivers[43,45,70,74] that may ultimately flow into the ocean increasing NDTI. Restricting direct livestock access to waterways can reduce sediment input[39]. Regenerative agricultural practices, such as rotational grazing, cover crops, and integrating trees, may reduce sediment production and export in pastures[26]. The absolute value of the effect size of natural vegetation was larger than that of pasture, suggesting that while converting grassland to another land use (ex. plantation) would decrease turbidity, planting native vegetation would gave even greater turbidity reduction benefits.

Similarly, the presence of a gravel road within a riparian zone increased NDTI (estimate: 0.032; 95% CI: 0.002 to 0.063; Fig. 2). Adding a gravel road crossing to a riparian zone that did not previously have one would have a similar effect on Gulf turbidity as increasing riparian pastureland by 39%. Additionally, gravel roads within entire watersheds increased turbidity in the IPW models without random effects (Supplementary Fig. 6). In contrast, paved roads did not have a statistically significant impact on turbidity, as observed in previous studies[75]. Because roads often co-occur with other degrative land use practices, such as agriculture and urbanization, the IPW models enabled us to isolate the specific effects of roads themselves. Although they cover a small land area, gravel roads have been found to have among the largest contribution to coral reef sediment of all land uses[23,26,75–78]. These findings are consistent with conditions in the region, where gravel road river crossings frequently lack bridges, resulting in vehicles fording streams. This practice contributes to elevated turbidity through several mechanisms, including sediment displacement, bank undercutting, and soil compaction leading to reduced infiltration[79]. Interventions to mitigate the impacts of gravel roads on coastal turbidity could include measures such as constructing bridges to prevent vehicles from fording rivers, avoiding steep slopes during road construction, and targeting high-risk segments for maintenance and erosion control[75,76].

We cannot reject the null hypothesis that plantations and exposed land have no effect on Gulf turbidity at either scale. The results did not show any significant influence of these land uses on turbidity, and the estimates associated with them had the widest 95% confidence intervals of any land use category (Fig. 2), suggesting considerable heterogeneity in their effects. This variability may arise because both the plantation category (encompassing oil palm, teak, and gmelina) and the exposed land category (including towns, beaches, and fallow agricultural land) represent diverse land use practices that may have divergent impacts on turbidity. Even within a specific practice, there may be heterogenous impacts on turbidity; for instance, sustainable small scale palm oil plantations may impact turbidity less than traditional plantations[60]. Additionally, since many land uses are correlated, we may have insufficient statistical power to detect the effects of the land uses. Previous studies have also found surprising relationships or no statistically significant relationships between urban land use and downstream ecosystem parameters[26,71]. Although no effect size was found, each of the specific land uses that comprise the urban and exposed land use category may have their own effect on Gulf turbidity.

## Turbidity impacts are detectable locally but not at the watershed scale

In contrast, we did not find evidence that land use or roads in the entire watershed significantly affected NDTI (Fig. 2, Supplementary Table S1). This finding carries two key implications for land-sea interaction research theory and methods. First, it underscores the disproportionately important role of riparian zone management over entire watershed management in influencing coastal and marine water quality. This supports the conceptualization of riparian zones and river corridors as critical linkages between terrestrial and marine systems, acting as primary conduits through which land-based activities affect downstream aquatic environments.

Second, it suggests that analyses relying solely on average watershed-scale land use, which is common in many land-sea studies, may overlook these localized but influential effects and underestimate the impacts of land use on marine ecosystems[3,15,24,47]. Our findings align with patterns observed in the well-studied Great Barrier Reef system, where vegetation degradation and loss of riparian cover drive disproportionate sediment and nutrient export[23,80,81], yet extend these insights by providing empirical, multi-decadal evidence in a data-limited tropical system.

Many of the mechanisms described in the previous section for how natural vegetation and pasture affect turbidity are even more impactful when those land uses are in the riparian zones. Natural vegetation in riparian zones provides localized benefits, such as streambank stabilization, increasing the complexity and biodiversity of river's hyporheic zones, and influx of organic matter to create natural dams[26]. Pastures in the riparian zone have an outsized influence on sediment loads because cattle grazing directly next to rivers and entering rivers to drink can exacerbate streambank erosion and mobilize sediments; for instance, Wilson & Everard (2018)[42] found a 90% increase in river turbidity due to riparian cattle trampling.

While gravel road presence within riparian zones increased turbidity in the gulf across all IPW models, gravel roads within entire watersheds were found to significantly increase turbidity only in the IPW model without random effects. Gravel roads can still elevate turbidity levels even when they do not directly intersect with rivers. Roads function as persistent linear disturbances across landscapes, and their contributions to erosion and runoff can originate from multiple points along their length. Additionally, roads can induce secondary geomorphic disturbances, such as landslides, which may substantially alter sediment transport and turbidity dynamics throughout a watershed[82].

Our results complement and extend previous studies in the freshwater ecology literature that have found that riparian land uses are more strongly correlated with river water quality than watershed land use[44–46,71,83,84]. For instance, Xu et al. (2023)[83] found that land use explained 41.4–50.3% of river water quality at the reach scale, and just 5.7–7.7% at the sub-catchment scale. In another Neotropical region, São Paulo, Brazil, Piffer et al. (2021)[45] found that the riparian zone scale better explained river turbidity than the watershed scale. Contrastingly, some previous studies have found larger impacts of land use on river water quality at the watershed scale[74,85,86].

## Land use impacted turbidity at least 800 m offshore

Riparian natural vegetation (largest effect was 400 m offshore; estimate −0.130) and pasture (largest effect was 600 m offshore; 0.098 estimate) both statistically-significantly influenced annual average turbidity at all distances 25–800 m offshore (Fig. 3, Supplementary Table S2). These results may be explained by the Gulf's oceanographic features and sediment dynamics. The Golfo Dulce contains extensive turbidite channels that transport sediment from river mouths into the deep inner basin, where turbidite accumulation rates are more than twice those along the outer slopes[87,88]. Additionally, the gulf's two-layer current system—a warmer, wind-driven surface layer and a colder, denser subsurface layer—may contribute to offshore sediment retention, especially where circulation is slower farther from the coast[67]. In contrast, gravel road presence in riparian zones only statistically significantly increased average annual Gulf turbidity 100–200 m offshore (largest effect was 200 m offshore; estimate 0.044). Gravel roads may be larger effects closer to the coast because previous studies have found that coarse sediments, such as those from unpaved roads, tend to settle near shore due to their larger grain size, while finer particles from other land uses are more easily transported offshore[13,20]. Future research could extend analysis beyond 800 m offshore.

## Forest conservation policy aligns with changes in land use and turbidity

Empirical land-sea interaction studies in the tropics linking land management policies to downstream marine outcomes across multiple decades remain scarce[10,23,89]. Because our analysis revealed that riparian and

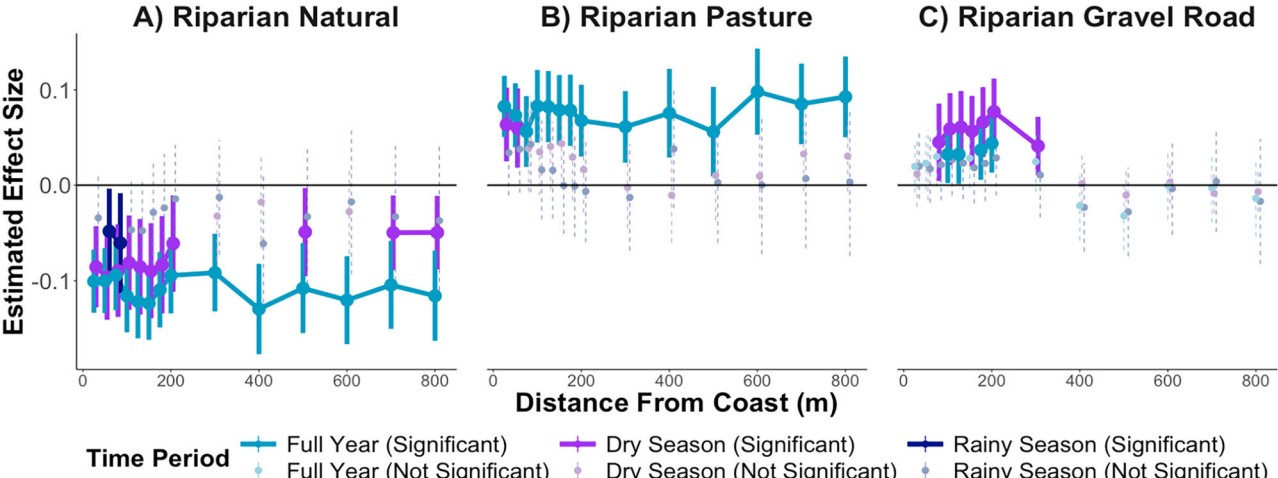

**Fig. 3 | Estimated effects of riparian land use on Gulf turbidity across seasons 25–800 m offshore.** Estimates from IPW models with random effects for the impact of **A** riparian natural vegetation cover, **B** riparian pasture cover, and **C** the presence of gravel roads within riparian zones on average gulf turbidity 25–800 m off the coast across the full year (blue), in the dry season (purple), and rainy season (dark blue). Significant models are shown in darker colors with solid 95% confidence intervals, and non-significant models are shown in lighter colors with dashed confidence intervals. Estimates are in Supplementary Table S2.

nearshore land use have a dominant influence on coastal turbidity, we examined how national forest conservation policies may have shaped land use in these areas over time. Our study period coincided with the implementation of major terrestrial conservation policies in Costa Rica, including the 1996 Forestry Law, which banned deforestation, established protections for riparian zones, and introduced a pioneering Payment for Ecosystem Services (PES) scheme. These interventions align with documented increases in regional natural vegetation (Fig. 4A, B; Supplementary Fig. 4) and decreases in pasture (Fig. 4C, D; Supplementary Fig. 4), both in riparian zones (Fig. 4A, C) and across full watersheds (Fig. 4B &D). We observed declining turbidity levels in the Golfo Dulce over the same time period (Fig. 4E, F). Similarly, following the 1990 designation of Piedras Blancas National Park, a shift from pasture, plantation, and exposed land to natural vegetation within the park (1987–1998) corresponded with reduced turbidity along adjacent coral-rich coastlines (Supplementary Fig. 5). While these trends cannot be conclusively attributed to policy alone—given concurrent drivers such as rising ecotourism, shifting public environmental values, and global declines in beef and palm oil prices[43,64,90,91]—the temporal alignment suggests these policies may have contributed. These findings link national-scale terrestrial policy interventions with local-scale improvements in coastal water quality, underscoring how land-based conservation initiatives—particularly those targeting riparian zones—can yield downstream marine co-benefits.

### Land use impacts on turbidity were greater near rivers that are shorter, steeper, and lower-discharge and in the dry season

We found that the effects of riparian land cover on coastal-marine turbidity varied depending on river characteristics (Fig. 5, Supplementary Table S3). The turbidity-reducing effects of riparian natural vegetation were greater near rivers with low discharge (interaction term estimate: 0.88) and high slope (interaction term estimate: –0.49). The turbidity-increasing effects of riparian grassland were greater near rivers with low discharge (interaction estimate: –1.34) and shorter rivers (interaction estimate: –0.17). Notably, we found no significant heterogeneity in the effect of gravel roads on turbidity, suggesting that their influence may be more consistent across systems or overshadowed by other factors. These interaction effects underscore that river morphology and hydrology mediate how land use influences turbidity. Thus, conservation strategies such as riparian restoration will likely yield the largest water quality benefits in shorter, steeper, and lower-discharge rivers, so these rivers may be prioritized in time and resource-scarce contexts. These impacts are likely to be especially pronounced when reforesting pastures, as this land use most increases turbidity, as discussed above. In

these systems, steeper slopes likely elevate erosion risk, and with shorter travel distances and less opportunity for sediment deposition, riparian land cover exerts a stronger influence on whether sediment reaches the coast.

Land use had stronger and more spatially extensive effects on turbidity during the dry season than the rainy season (Fig. 3). There are a few potential reasons for this. This attenuation[84] might be related to the fact that the heavy rainfall may produce similar runoff volumes across land cover types, and overwhelm the buffering capacity of, e.g., forest litter and vegetative barriers, thus increasing sediment loads regardless of land use[84]. Rainy-season turbidity may be more strongly governed by non-land-use factors such as slope, catchment area, and localized rainfall variation. Additionally, a previous study in the Golfo Dulce found different wind directions in the rainy and dry seasons; in the dry season, wind partly reduced freshwater flow to the surface layer of the gulf, while wind may strengthen freshwater currents in the dry season[92]. Cloud cover during the rainy season reduced the number of NDTI observations, reinforcing the focus on mean values that capture persistent spatial patterns. Similarly, land use was more predictive of water quality in the dry season than the rainy season in studies in China[84,93]. Overall, these patterns suggest that while extreme rainfall events may dominate total sediment export, the chronic, baseflow effects of land use on turbidity are most evident during the dry season, highlighting that climatic drivers are less dominant then, and management interventions may be more effective.

### Comparing causal analysis approaches

We compared a suite of six additional modelling approaches in addition to the IPW models discussed above, each with different assumptions, to evaluate the robustness of our results: naïve linear model, mixed-effects model, two-way fixed effects, IPW without the random effect and with clustered robust standard errors, group mean covariate, and group mean centered. The consistent direction of effects—negative for riparian natural vegetation and positive for riparian pasture—across model types (Fig. 6; see Supplementary Note 2 for a more thorough discussion of model assumptions) reinforces evidence for a plausible causal relationship.

Sensitivity analysis, testing the robustness of our estimates to potential omitted variable bias[94], suggests that the statistically significant treatment effects of the IPW models with random effects were robust to reasonable levels of unobserved confounding (Supplementary Fig. 7). Together, these results suggest that key modelling assumptions were not substantially violated (see Supplementary Notes 1 & 2 for more details). Remotely sensed NDTI statistically significantly predicted the *in situ* Nephelometric Turbidity Units (NTU) measurements (Supplementary Fig. 9; see Supplementary Note 1 for more details).

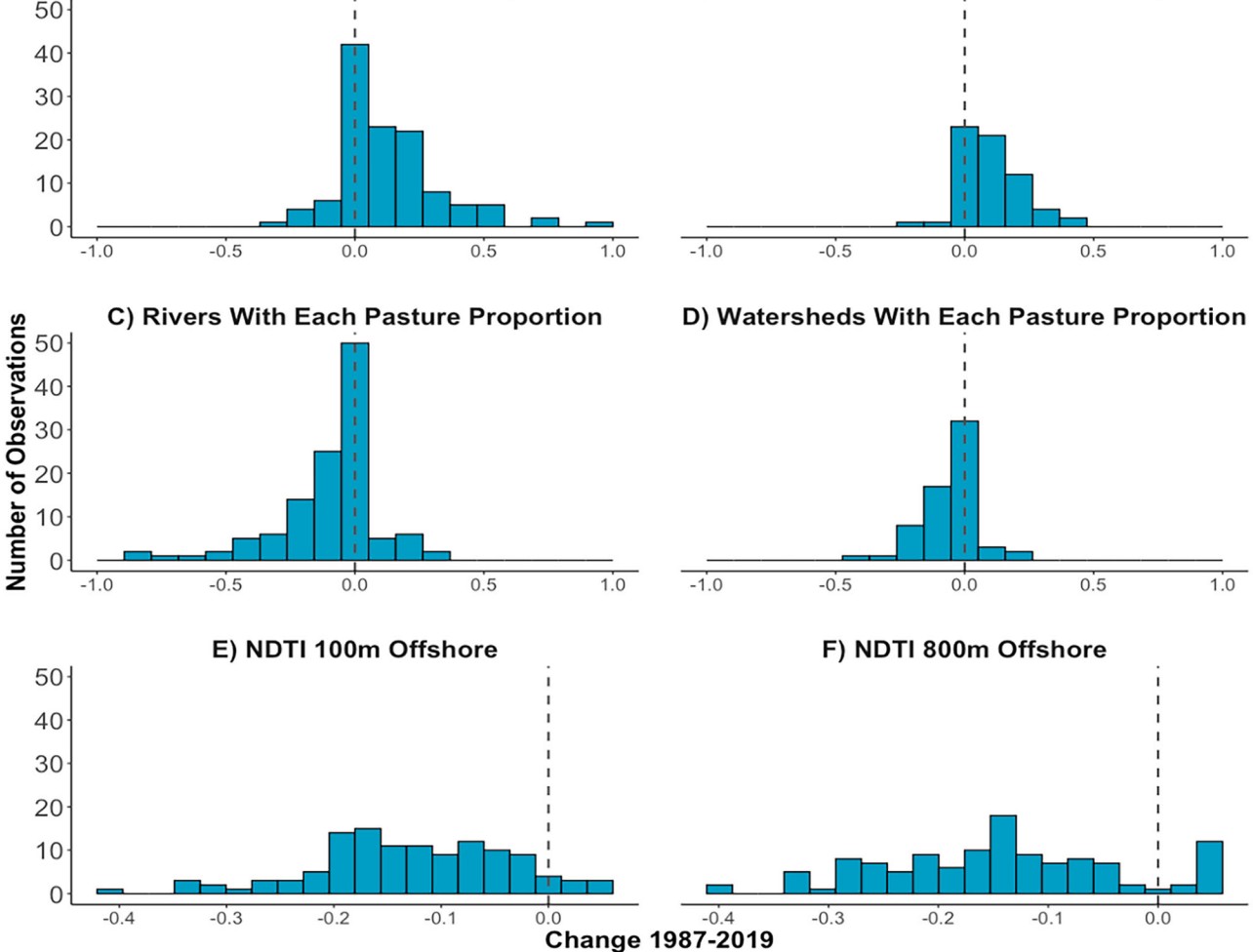

**Fig. 4 | Land use and turbidity change over the study period.** Over the study period (1987–2019), quantity of **A** rivers with each change in proportion of natural vegetation cover in their riparian zones, **B** watersheds with each change in proportion of natural vegetation cover, **C** rivers with each change in proportion of riparian pasture cover in their riparian zones, **D** watersheds with each change in proportion of pasture cover, **E** points 100 m offshore of river mouths with each change in NDTI value, and **F** points 800 m offshore of river mouths with each change in NDTI value.

## Implications for marine ecosystems

The observed reduction in turbidity associated with riparian natural vegetation likely has direct implications for downstream ecosystems. In the Golfo Dulce, both coral reefs and seagrass meadows are concentrated within the first few hundred meters offshore[68,95–98], the same region where turbidity is most affected by land use (Fig. 1). Elevated turbidity from pasture and gravel roads likely causes chronic reductions in light penetration and increased sediment deposition, consistent with mechanisms shown to diminish coral health, ecosystem services, and resilience to climate change[1–4,48,56,89]. Chronic exposure to elevated turbidity, such as from land use drivers like pasture and gravel roads that we evaluated in this study, may be more detrimental to coral health than short-term pulses[99–101]. All known coral reef sites in the Golfo Dulce are located within 800 meters of the coast and near river mouths, placing them directly with the influence of land-derived turbidity[66–68]. Field studies in the Golfo Dulce have recorded high rates of terrigenous sediment accumulation, sediment trapped in coral tissue, and stable isotope ratios indicating strong riverine influence on reef geochemistry and ecology[68,88,98,102]. Because baseline turbidity in Golfo Dulce is generally low during the dry season, even modest increases in turbidity are likely to have meaningful ecological consequences[103]. Some coral reefs in the region have increased in their live coral coverage over the last few decades[68,97,98]. This timeframe aligns with the observed increases in natural

vegetation and decreases in turbidity in the region (Fig. 4, Supplementary Fig. 4 & 5). As we discuss in detail in Supplementary Note 3, it is not possible to fully attribute the increase in live coral cover to these land use and turbidity changes due to the limited existing coral reef monitoring and the large number of factors that may affect coral reef health, but it is possible that these factors may have contributed based on the spatial overlap and previous research establishing the relationship between turbidity and coral health[2,4,48,56,89,104].

Similarly, seagrass meadows in Golfo Dulce are located within the shallow nearshore turbidity zone, and previous research shows that light limitation is a key constraint on their depth distribution and persistence[96]. High turbidity and nutrient enrichment reduce light availability, restricting seagrasses to shallower depths and decreasing their health and primary productivity[21,96]. This in turn reduces their capacity for various ecosystem functions and services, including carbon storage, sediment stabilization, and nutrient removal[21,96]. Previous studies have found that land use was a strong predictor of tropical seagrass condition[105]. Urban, industrial, or agricultural runoff is one of the most ubiquitous threats to seagrass species in the region[106].

Improved water quality and reduced sedimentation have been shown to enhance coral and seagrass resilience to climate change and other stressors[2–4,106]. For instance, reefs in Hawaii with reduced land-based

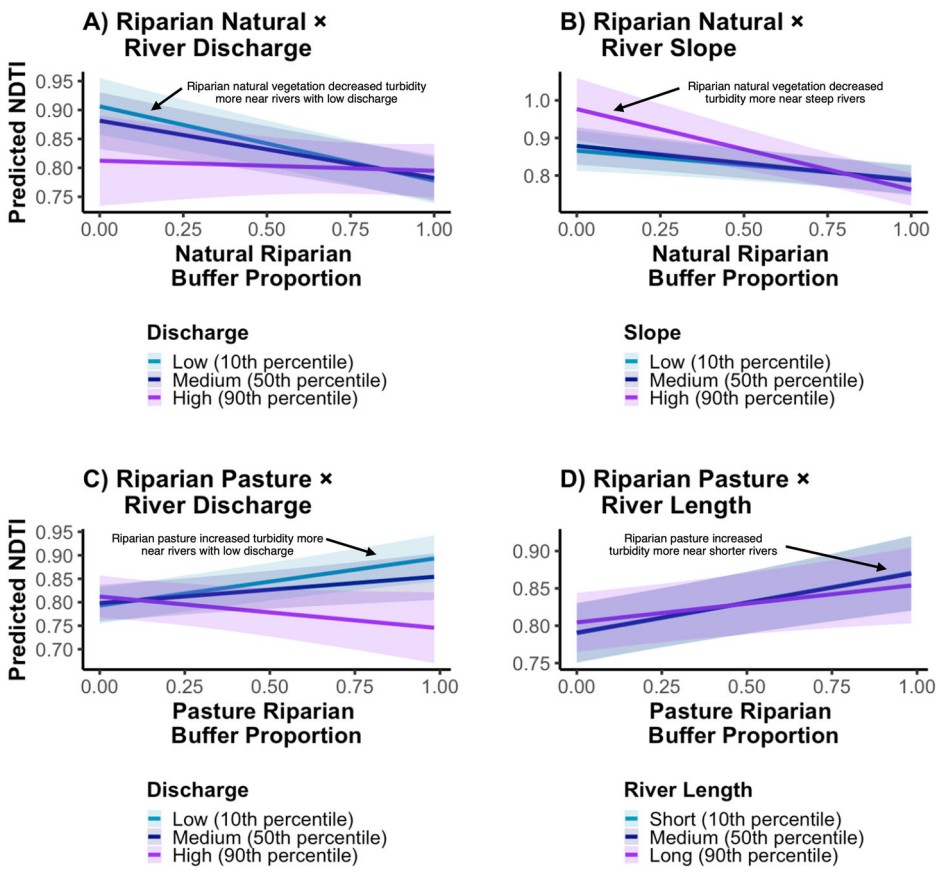

**Fig. 5 | Interaction effects of riparian land use and river covariates on predicted turbidity.** Relationships between the proportion of natural vegetation in riparian zones and NDTI, shown for rivers with low, medium, and high (10th, 50th, and 90th percentiles, respectively) discharge (**A**) and slope (**B**). Relationship between the proportion of pasture in riparian zones and NDTI, shown for rivers with low, medium, and high discharge (**C**) and short, medium, and long length (**D**). Predictions are from inverse probability weighted (IPW) regression models with covariate adjustment weighted for the treatment variable. Shaded areas indicate 95% confidence intervals. Figures illustrate statistically significant interaction effects; see Supplementary Table S3 for estimates for all combinates of river covariates for the statistically significant land uses.

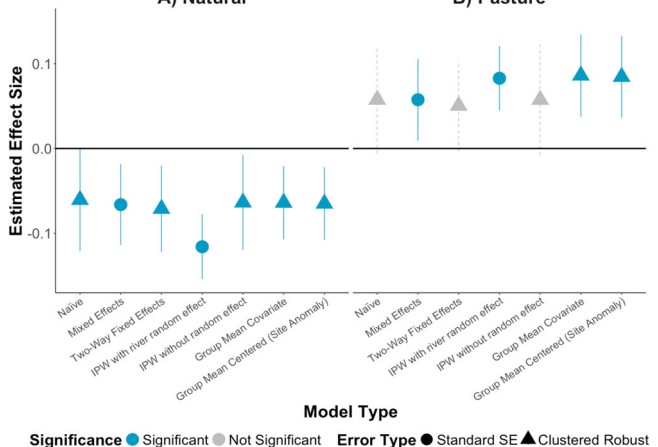

**Fig. 6 | Comparison of estimates between modelling approaches.** Estimates of the impact of increasing riparian natural vegetation (**A**) and pasture (**B**) on NDTI 100 m off the coast based on the naïve linear model, mixed-effects model, two-way fixed effects, IPW with the random effect for river, IPW without the random effect, group mean covariate, and group mean centered. The group mean centered estimate is for watershed land use anomaly. Blue indicates a p-value of the estimate less than 0.05, and gray indicates a p-value greater than 0.05. Circles indicate models including standard SEs and river-level random effects, and triangles indicate models with clustered robust SEs and no river-level random effects. Bars represent the 95% confidence intervals.

impacts exhibited lower coral mortality following severe heat stress[2], and modeling in Madagascar found that the effects of deforestation on reef sedimentation outweighed those of future climate change[3]. Likewise, studies on the Great Barrier Reef show that maintaining low nutrient and sediment

inputs can substantially improve coral tolerance to heat stress[4], and Samper-Villareal et al. (2025) noted that natural disturbances do not typically substantially threaten seagrasses in the absence of anthropogenic pressures[106]. These findings suggest that riparian restoration and land use management, which we found to reduce turbidity, could play a meaningful role in supporting coral and seagrass resilience to disturbances, such as climate change.

The land-use-driven changes in turbidity we document likely have cascading implications not only for coral reefs and seagrasses, but also for associated biodiversity, fisheries, and blue-carbon ecosystem services[19,105,107]. Although our analysis does not establish direct causal effects on marine ecosystems, the spatial overlap between turbidity changes and the distribution of coral and seagrass habitats strongly indicates potential downstream consequences, such as for reef-associated fishes and invertebrates, small-scale fisheries, tourism, and coastal protection[19,105,107]. These results underscore how riparian management can indirectly bolster coastal ecosystem health and resilience by maintaining water clarity, a key buffer against the accelerating effects of ocean warming and other global stressors[2,4,108–110]. Future research could collect more *in situ* turbidity, sea grass, coral reef health, and other marine ecosystem data to study these impacts more directly.

**Implications for future land-sea interaction research**

This study employs a replicable methodology that integrates freely-accessible satellite imagery, public databases, and causal inference techniques to assess land-sea interactions across large spatial and temporal scales, particularly in remote or data-limited regions. Specifically, this framework advances land-sea research in four main ways. First, whereas most existing studies rely on correlational statistical analyses[5,26,54] that cannot isolate the specific effects of land use from co-varying biophysical conditions, we explicitly account for confounding factors (ex. slope, soil type, and other land uses) using causal inference. Second, although catchment and hydrodynamic models offer valuable mechanistic insight into sediment export

and offshore dispersal, they typically require detailed calibration and validation datasets that are unavailable in many tropical systems, and are rarely validated against long-term observed water quality outcomes[10,23,34,57,60,61]. Recent work couples terrestrial catchment/erosion models with hydrodynamic or dispersal models (ex. plume transport, nearshore circulation) to more explicitly link land-based sources to downstream marine exposure, but their application remains constrained in data-limited contexts by parameterization and validation requirements[25,34,37]. We complement these process-based approaches by providing long-term, observation-based validation of sediment exposure patterns using multi-decadal satellite records. This empirical framework allows us to test, validate, and generalize patterns predicted by process-based modeling studies using real-world observations.

Third, much of the land-sea interaction and ridge-to-reef management literatures focus on a few watersheds or rivers[13,20,26,54,75,111]. Ground-based studies are not scalable, which can limit generalizability, obscure spatial heterogeneity, and reduce relevance for decision-making[6,23,24,26,31,33]. The use of remote sensing allows us to analyze many catchments simultaneously while leveraging temporal replication to separate persistent patterns from river-specific variability. This scalability is especially important in data-limited regions, where publicly available data may be the only feasible evidence base for local decision-making. Fourth, we show that land-sea relationships are spatially scale-dependent, both terrestrially (riparian vs. watershed extents) and marine (distance offshore). Many previous land-sea assessments assume a single relevant spatial scale[3,24,25,47], which may obscure effects that emerge only in narrow riparian buffers, as have been well-documented in the freshwater literature. Our results demonstrate that different land uses became detectable at different specific spatial scales, underscoring the importance of explicitly considering spatial heterogeneity to better capture the full extent of ecological linkages. Together, these contributions provide empirical evidence that strengthens and complements process-based approaches in data-limited coastal systems.

Future research could integrate our empirical approach with process-based catchment or hydrodynamic models to improve predictions of sediment fate and ecological impacts. In particular, coupling empirical turbidity estimates with oceanographic dispersal modeling could help resolve fine-scale variation in exposure among coral, seagrass, and mangrove habitats.

To improve precision, future research could incorporate in situ and higher-resolution remote sensing data. Field-based measurements of river discharge, turbidity, and riparian characteristics (e.g., canopy cover, species composition) could help validate remotely sensed data, improve model accuracy, and support assumptions required for causal inference. While field data typically cover smaller spatial extents than satellite imagery, they can be compared with landscape-scale analyses to assess consistency. Technologies such as unmanned aerial vehicles (UAVs) and LiDAR also offer promising avenues for collecting fine-scale, spatially extensive data on land use, vegetation structure, and water quality. In areas lacking detailed land use maps, researchers may also consider vegetation indices like NDVI or EVI, which have shown strong inverse correlations with marine turbidity in other systems[112].

## Limitations

In addition to limitations discussed throughout, there were several other limitations to this study. First, while NDTI is useful for estimating relative turbidity at large scales and strongly correlated with *in situ* NTU, it is not a direct measurement of exact *in situ* turbidity values. Second, we used average annual precipitation from 1960-2013 in the river discharge calculation, though interannual variation in precipitation—and therefore discharge—likely occurred over the study period. Third, the land use maps did not capture high-impact events with small spatial footprints, such as construction of roads and large buildings, that could substantially influence Gulf turbidity and coral reef health over short periods of time. Finally, there are other factors that may affect turbidity in the Golfo Dulce that were not captured in the model covariates, which may have increased the standard

errors. For instance, additional factors, including coastal circulation (beyond the counterclockwise current, which we account for), turbidite channels, salinity, tides, wind, proximity to the open ocean, could influence turbidity beyond the variables modelled here[87,88,113]. The impact of riparian land use on Gulf turbidity may also be mediated by other river-specific variables not captured in the models (ex. channel morphology, stream order, morphology modification due to urban development[74,84,113]).

## Riparian zone conservation as a potential win-win-win for terrestrial, freshwater, and marine conservation

Our results indicate that conserving and restoring riparian buffers offers a potential "win-win-win" for terrestrial, freshwater, and marine ecosystems. This may be particularly beneficial where full watershed conservation is unfeasible due to competing land-use demands, such as those driven by livestock production. Riparian buffer conservation and restoration are rare interventions that consistently advances multiple conservation goals and offers high conservation returns on investment[27,29,47,89]. Our results suggest that in contexts where time and resources for terrestrial management are limited, restoring natural vegetation in riparian zones—particularly those currently dominated by pasture and located along short, steep, low-discharge rivers—will yield the greatest reductions in marine turbidity. In addition to the marine ecosystem benefits discussed here, riparian buffers support terrestrial objectives—such as biodiversity, ecosystem functioning, and the provision of ecosystem services—and freshwater goals like aquatic biodiversity, water quality, and aquatic ecosystem services[114,115]. Moreover, riparian conservation and restoration may deliver social and economic co-benefits: they improve forest-based livelihoods and water quality for upstream communities, while enhancing reef health, coastal fisheries, and tourism revenue downstream[24,114]. Existing policy frameworks in many countries, including Costa Rica, already mandate riparian buffer protection, potentially easing implementation and enforcement. Stricter enforcement of the Costa Rican Forest Law protecting riparian buffers, along with more targeted restoration in degraded riparian areas, will likely help buffer sedimentation threats to coastal marine ecosystems. Mangroves in estuaries also provide a final line of defense by trapping sediment before it enters marine ecosystems; mangrove restoration could therefore be especially impactful[7,17,26,28,30,32,63,73,89,116–119].

Recent work has called for integrated seascape restoration that links connected marine habitats such as coral reefs, mangroves, and seagrasses[63,120]. Our results suggest that this framework could be expanded upstream to explicitly include riverscapes, where riparian forest and mangrove restoration can enhance water quality and strengthen the resilience of adjacent marine ecosystems. Incorporating riparian restoration into seascape and blue carbon initiatives could better capture the full suite of land-sea linkages that sustain coastal biodiversity and ecosystem services. Strengthening the link between terrestrial and marine ecosystem services may unlock opportunities for additional innovative fundings and financing flows. In areas like the Osa Peninsula, where tourism depends on clear waters and healthy coral reefs, investments in riparian restoration could generate direct economic returns through improved coastal water quality.

Spatial prioritization studies highlight the challenge of achieving cross-realm objectives, as priorities often shift depending on the focal ecosystem[27,29,47,89]. More broadly, our findings offer empirical evidence of the cascading impact of land use on marine ecosystems and underscore the importance of integrated terrestrial, freshwater, and marine conservation planning, despite the challenges of cross-ecosystem coordinated management, including ecological-political boundary mismatches, fragmented governance, conflicting stakeholder priorities, and knowledge gaps[26,28,32,89,119]. Our findings suggest that marine-focused stakeholders may find strategic value in engaging in terrestrial policy processes, while land-based conservation initiatives could strengthen outcomes by accounting for downstream marine impacts. Riparian restoration and terrestrial protected areas could be coordinated with marine-focused interventions, such as Marine Protected Areas (MPAs), to amplify ecosystem resilience, particularly where MPAs are hydrologically linked to forested upstream

watersheds[26]. Where coral reef protection is the primary objective, targeted marine strategies—such as coral restoration, thermal shading, or microbial therapies—may offer greater immediate returns than ridge-to-reef approaches, though they lack terrestrial co-benefits and face implementation challenges[30,117,118]. Decision-support tools can help guide when to act on land; for example, a decision tree has been developed to help practitioners, resource managers, fishers, and other resource users determine whether land use should be considered to achieve coastal marine conservation goals[119]. In our system, that decision tree indicates that land-based drivers are critical, consistent with our results. Lessons from systems where ridge-to-reef management frameworks are already well developed may also be transferable to similar contexts. For example, the Great Barrier Reef benefits from long-standing policy instruments such as the Reef 2050 Plan and the Reef Water Quality Protection Plan, as well as the eReefs interoperable platform that integrates monitoring, hydrodynamic-biogeochemical modeling, and decision-support tools[10,23,37,81,121].

## Conclusion

Our results underscore the role of riparian conservation and restoration as a strategic lever for ridge-to-reef sustainability and demonstrate the potential for coordinated land-freshwater-marine strategies to deliver measurable ecological and societal benefits in a changing world. We provide one of the first large-scale empirical analyses of how riparian and watershed land uses affect coastal turbidity, bridging a long-standing gap between fine-scale field studies and broad-scale modeling efforts. Using a scalable, observation-based framework that integrates multi-decadal satellite-derived turbidity data with causal inference models, we disentangled the effects of land use, geomorphology, and policy on sediment export across multiple tropical watersheds. Natural vegetation reduced coastal turbidity, while pasture and gravel roads increased it, with effects extending at least 800 m offshore and strongest at the riparian scale. Areas where turbidity was most affected by land use overlapped with critical coral reef and seagrass habitats, implying potential downstream consequences for biodiversity and ecosystem services. Turbidity also declined alongside increases in riparian vegetation following implementation of Costa Rica's Forestry Law and Payment for Ecosystem Services program, suggesting that terrestrial conservation policies can yield measurable marine co-benefits. Together, these findings identify reforesting or conserving riparian zones, particularly in pasture-dominated, short, and steep rivers, as important interventions at the land-sea interface. More broadly, our results show how empirical, scalable evidence can strengthen cross-realm conservation planning, inform policy and investment to target the most effective actions, and advance the scientific basis for linking terrestrial management to downstream ecosystem resilience. This work underscores the need for integrated ridge-to-reef frameworks that align conservation, restoration, and financing across landscapes, riverscapes, and seascapes.

## Methods

### Study area

The Osa Peninsula, situated on the southern Pacific coast of Costa Rica, is rich in biodiversity and harbors the most extensive remnant of lowland wet forest in the Pacific Mesoamerican region (Fig. 1)[66,68,122]. The Osa Peninsula is 85% forested, the majority of which is tropical wet forest, and holds premontane wet and tropical moist forest types[64,123]. The peninsula's altitude varies from sea level to over 775 meters[124]. It receives an average annual rainfall of over 5,000 mm, which mostly falls during the wet season from May to November, peaking August-November[125].

The Osa Conservation Area (ACOSA), one of 11 conservation areas in the Costa Rican National System of Conservation Areas (SINAC), encompasses the Osa Peninsula and extends onto mainland southwestern Costa Rica across the Golfo Dulce. All the rivers that flow into the Golfo Dulce are in ACOSA. Rivers from three ACOSA protected areas flow into the Golfo Dulce: Corcovado National Park on the Osa Peninsula, established in 1975[126]; Piedras Blancas National Park, located on the Costa Rican mainland, established in 1990[127]; Golfo Dulce Forestry Reserve,

which stretches from the peninsula to the mainland and was established in 1978[126].

Agricultural expansion for cattle, bananas, and oil palm degraded ACOSA's tropical rainforest in the mid-twentieth century[128]. Much of the forest has regenerated in recent decades, likely due to a combination of the creation of protected areas, the Costa Rican Forestry Law of 1996 (which banned deforestation), the development of a pioneering Payment for Ecosystem Services program, an increase in ecotourism and public environmental interest, and a global decrease in beef and palm oil prices[64,90,91,129]. Ecotourism, cattle ranching, palm oil production, and artisanal fishing are the main economic activities in the region. Osa's riparian zones are particularly degraded[64], and riparian land use in the region has been shown to affect river water quality for at least a kilometer downstream[43].

The Golfo Dulce, one of four tropical fjord-like formations on Earth, separates the Osa Peninsula from mainland Costa Rica (Fig. 1). Like the Osa Peninsula, the Golfo Dulce is extremely biodiversity rich, including fish, sea snakes, coral reefs, seagrasses, plankton, and mollusks[65–67,96,122,130]. The gulf reaches a maximum depth of 201 meters and is separated from the Pacific Ocean by a shallow sill[87,88,131].

Coral reefs in the Golfo Dulce died at very high rates in the 20th century, and previous studies have suggested that terrestrial land activities may be a driver, in addition to natural disasters and El Niño/Southern Oscillation (ENSO) events[68,97,132]. Local coastal communities consider sedimentation and agrochemicals to be among the main threats to coral reefs in Golfo Dulce[133]. The effect of riparian land use on oceanographic conditions and coral reef health has not been analyzed in this remote tropical region[134].

### Overview of modeling approach and data

We used a suite of causal inference models to try to disentangle the impacts of land use on Gulf turbidity at different spatial scales. There are many confounding factors that may affect any given land use in the region and turbidity in the Golfo Dulce, which we identified through a literature review and experience in the region[26,56,64,65,68,88,135]. We examined the effect on turbidity by the percentage of land area covered by natural (mature forest, secondary forest, mangrove, wetland, and water), pasture, plantation (oil palm and teak/gmelina), and exposed (urbanization, beach, unused/fallow agriculture) and the presence of gravel and paved roads within watersheds and riparian zones. Riparian zones were 15 m wide because this is the width protected under the Costa Rican Forestry Law 7575 of 1996 in rural areas, and because recent research in the region has suggested that land use in 15-meter-wide buffer zones is more strongly correlated with downstream river water quality than land use in wider riparian zones[43].

For each treatment land use, the confounders include other land uses (including roads), the proportion of that land use in other parts of the watershed, slope, soil type, and other unobserved confounders. For instance, the impact of riparian natural vegetation may be confounded by other land uses in the riparian zone such as roads or oil palm plantations in the same riparian zone because these land uses both affect the treatment–natural vegetation proportion–and also independently impact the response—gulf turbidity[26]. Natural vegetation within the full watershed may also affect both land use in the riparian zone and gulf turbidity. Geomorphological parameters such as river slope and soil order may also confound the relationship because they both affect land use patterns and properties that affect turbidity such as erosion rates and soil mean particle size[26,64,84,136].

The main datasets used in this analysis were derived from freely available NASA satellite imagery, Costa Rican open-source datasets, and in situ data collected by collaborators. Data extraction was conducted in Google Earth Engine (GEE), Python, and QGIS, and data analysis and visualization were conducted in R.

### Land use and road data

We generated a map of rivers that flow into the Golfo Dulce by compiling river layers from the Digital Atlas of Costa Rica 2014[137], the recent orthophoto and cartography initiative from the Costa Rican National Registry[138],

and manually digitizing rivers based on topography and satellite imagery. In total, we identified 119 unique river mouths across 64 watersheds that flow directly into the Golfo Dulce. We considered river mouths to be unique if they were at least 300 meters apart; consequently, some rivers have multiple river mouths. We created a buffer of 15 m on either side of each river to delineate the riparian zones.

We extracted land use and land cover (LULC) in 1987, 1998, and 2019 in each of these polygons from recent LULC maps created for the study area at 30 m resolution[64]. The land use maps were created by processing the Surface Reflectance Tier 1 Landsat 5 Thematic Mapper, Landsat 7 Enhanced Thematic Mapper Plus (ETM + ) and Landsat 8 Operational Land Imager (OLI) with the GEE API and training a supervised, pixel-based random forest machine learning algorithm using an 80/20 split for training and validating ground truth data polygons. The LULC maps classified forest, palm plantations, mangroves, water, grasslands (which are most livestock pastures in this region), urban areas and exposed soils, wetlands, and teak and gmelina plantations as the dominant LULC classes in the region and had 90.44% accuracy.

Portions of two of the 88 sub-watersheds that flow into the Golfo Dulce extend past the boundaries of previous LULC for the region[64]. Consequently, we expanded the maps to encompass the full extent of these watersheds. Using the same methods as Brumberg et al. (2024), we collected imagery to cover these areas of expansion. We investigated landcover in these areas using a support vector machine (SVM) learning algorithm in ArcGISPro Version 3.1.4. We collected new ground truth polygons for training data to identify parameter values for each class in the map extension. The SVM classifier does not require the samples to be normally distributed, and it is less susceptible to noise and an unbalanced number or size of training sites within each class. Training data were collected based on landscape trends in spatial patterns and distributions as seen by the naked eye and by spectral signatures. Following the same schema, this output a classified raster of dominant LULC classes in the boundary extension.

We extracted the percentage of each LULC class in each watershed and riparian zone in 1987, 1998, and 2019 using the *pandas* and *geopandas* libraries in Python. Land uses that were expected to have similar effects on coastal turbidity were aggregated: new "natural" (mature forest, secondary forest, mangrove, wetland, and water) and "plantation" (palm and teak/gmelina) classes were created, as discussed above.

Presence of gravel and paved roads within each watershed and riparian zone were identified using a roads shapefile from the National Geographic Institute of Costa Rica[138]. Gravel and dirt roads were aggregated into a gravel road category. We created a function to extract gravel and paved road overlap with each watershed and riparian zone using the geometry module from *shapely* and *geopandas* in Python. Historical road data were not available, and it is possible that many had not yet been constructed in 1987 or 1998.

### Additional river characteristics: Soil type, length, slope, and discharge

We determined the dominant soil type in each watershed and riparian zone based on soil order and suborder shapefiles from the University of Costa Rica[139]. River length was calculated by summing the lengths of all river reaches that flow into each river mouth. Then, the slope of each river was estimated by extracting the highest elevation of any reach of the river from the 30 m Advanced Spaceborne Thermal Emission and Reflection Radiometer (ASTER) Global Digital Elevation Model (GDEM) Version 2[124] and dividing the elevation by the length of the river reach.

We used the Rational Method to estimate river discharge since we did not have *in situ* discharge data for each of the 147 rivers. The equation for the Rational Method is[140]:

$$\text{Discharge} = P * C * A \qquad (1)$$

where *P* is the average rainfall depth, *C* is the runoff coefficient, and *A* is the catchment area. For our research question, it was not necessary to determine absolute values of discharge but rather relative discharge between each river, so we used proxies for some variables when data was not available. For *P*, we calculated the average annual rainfall in each riparian zone based on a shapefile of average annual precipitation 1960-2013 from the Costa Rican National Meteorological Institute[125]. We calculated weighted averages of the annual rainfall in each riparian zone by intersecting the riparian zone polygons with the precipitation polygons, computing the weighted precipitation for each intersecting area, and summing them. The runoff coefficient *C* is usually heavily based on slope, so we used slope for the coefficient. Because there are multiple rivers in each watershed and each river likely has different discharge, we used river length instead of catchment area for *A* because they are highly correlated[141].

### Gulf turbidity and depth

Due to limited access to field data in this remote tropical region, this study relied on remotely sensed proxies for key variables. Exclusive use of remote sensing introduced some loss of accuracy and precision in model estimates. However, this approach also constituted a major strength, allowing us to analyze numerous rivers over a 30-year period and better account for spatial and temporal confounding.

The Normalized Difference Turbidity Index (NDTI) uses optical measurements obtained through remote sensing to quantify water turbidity. It integrates red and green spectral bands to determine relative turbidity (Eq. 2)[142,143]. Clear water exhibits higher reflectance in the green band compared to the red band, and reflectance of the red band increases as turbidity increases. Values range from -1 to 1, with higher values indicating higher turbidity and lower values indicating lower turbidity. NDTI is often used to compare relative turbidity, such as differences between groups or changes over time[142,144]. This is a scalable metric for our research question to understand relative spatial temporal patterns in turbidity over the entire Golfo Dulce and applicable to other data-scarce and remote regions[142].

$$\text{NDTI} = (\text{Red} - \text{Green})/(\text{Red} + \text{Green}) \qquad (2)$$

The GEE API was used to collect and process Surface Reflectance Tier 1 Landsat 5 TM, Landsat 7 ETM + , and Landsat 8 OLI data. Functions were created to mask cloud pixels in Landsat 5, 7, and 8 imageries using the Quality Assessment (QA) bands. Annual average NDTI, average rainy season NDTI (August-November), and average dry season NDTI (January-April) were calculated. This approach follows previous studies (e.g., Brown et al., 2017) showing that mean turbidity values effectively represent spatial differences in sediment export while minimizing the influence of episodic peaks.

River mouths are ideal sites for assessing turbidity impacts from riparian land use, as they represent the initial marine zones influenced by fluvial inputs, making them critical interfaces where terrestrial stressors first manifest in coastal-marine ecosystems[36]. To determine how far off the coast that land use affected gulf turbidity, turbidity was extracted at 14 distances perpendicular from the coast for each river mouth in each year and season. For each river mouth, points were manually drawn in QGIS 25, 50, 75, 100, 125, 150, 175, 200, 300, 400, 500, 600, 700, and 800 meters offshore of the coast. These distances encompass the range within which nearly all coral reefs and seagrass beds in the Gulf occur, based on georeferenced habitat coordinates compiled from peer-reviewed publications, Costa Rican government reports, and unpublished regional data[68,95–98]. A total of 1666 points were included, comprising 14 distances for 119 rivers. NDTI values were extracted at each point in the years corresponding to the land use maps (1987, 1998, and 2019) to enable a multi-decadal analysis impacts of land use on NDTI.

The Golfo Dulce has a counterclockwise current[68]. Gulf turbidity at the mouth of a river may be carried by this current counterclockwise, affecting the turbidity at down-current river mouths. To better disentangle the impact of each river's land use on gulf turbidity and to try to control for the currents, we included the gulf turbidity at the same distance offshore and in the same time period from the most proximal up current river as a covariate in all models.

The depth of the gulf (which ranges from 0–201 meters) may affect sediment and other particulate transport. The depth at each of the 1666 gulf sampling points was extracted from a bathymetry raster from the University of Costa Rica[131] using *rasterio, geopandas*, and *numpy* libraries in Python. Since the bathymetry raster does not extend all the way to the gulf coast, some points right next to the coast did not overlap with the bathymetry layer. Missing data values were imputed with -2 because this was the most common depth in the bathymetry raster within 400 meters off the coast.

## Modelling with inverse probability weighting

The main model type we used was generalized linear mixed-effects models with Inverse Probability Weighting (IPW). Generalized additive mixed models (GAMMs) indicated approximately monotonic relationships between land use and NDTI across most of the observed range (Supplementary Fig. 3). Based on this, we proceeded with linear models for the main analysis, assuming linearity was a reasonable approximation. Generalized linear mixed-effects models were used because they allow for random effects, account for additional structure in the dataset, and handle non-normally distributed data. IPW is a causal inference method that accounts for confounders that interfere with our ability to use regression to assess the relationship between a treatment and outcome[59,145]. We chose IPW over matching due to the small sample size and small area of common support between covariates. IPW requires the assumptions of no unmeasured confounding, positivity, no interference, consistency, compliance, and correct specification for causal interpretation (see Supplementary Note 2 for discussion of these assumptions)[145–149].

Separate models were run with each combination of a) treatment land use variable (proportion natural vegetation, proportion pasture, proportion exposed/urban land, proportion plantation, presence of gravel roads, and presence of paved roads), b) treatment spatial scale (watershed and riparian zone), and c) turbidity response in i) each distance offshore (25, 50, 75, 100, 125, 150, 175, 200, 300, 400, 500, 600, 700, and 800 m) and ii) each time period (annual, rainy season, dry season), for a total of 504 models. For each model, different weights were generated based on the specific confounders for that treatment. Observed confounders were other land uses at that spatial scale, the percentage of the treatment of land use in the whole watershed, presence of gravel and paved roads, slope, and dominant soil order. Weights were generated using the WeightIt R package version 1.4.0 by calculating weights in 1987 for each river and applying it for all three years[150]. When extreme weights were generated, we trimmed weights above the 97th percentile using the WeightIt *trim* function[150]. The full models included the treatment, other land uses at that spatial scale, the treatment land use at the watershed scale, gravel road presence, dominant soil order, river discharge, up current gulf turbidity, depth of the gulf where turbidity was measured, year, and river random effect (see Eq. 3 for example model structure for the treatment riparian natural vegetation). For each model, land use was dropped to avoid including highly correlated land uses in the same model and to ensure that land use proportions did not add up to one. Since each site was sampled in 1987, 1998, and 2019 for land use and turbidity, year was a fixed effect. The estimate, standard error, and *p*-value were extracted for the treatment in each model and compared. Since the roads shapefile was only available for 2018, models where roads were the treatment were only run for the last time period (2019). Models where the other land uses were the treatment and roads were a confounder extrapolated roads for all three years. To further examine the effects of different natural vegetation classes, sub-analyses were also conducted with each natural vegetation class (i.e., secondary forest, mature forest, mangrove, and wetland) as the treatment.

$$
\begin{aligned}
NDTI_{it} = {} & \beta_0 + \beta_1 \cdot Natural_{river}\_it + \beta_2 \cdot Natural\_watershed\_it \\
& + \beta_3 \cdot Palm_{river}\_it + \beta_4 \cdot Teak_{river}\_it + \beta_5 \cdot Exposed_{river}\_it \\
& + \beta_6 \cdot GravelRoad_{river}\_it + \beta_7 \cdot SoilOrder\_i + \beta_8 \cdot Depth\_i \\
& + \beta_9 \cdot Discharge\_i + \beta_{10} \cdot UpcurrentNDTI\_it + \beta_{11} \cdot Year\_t \\
& + u\_i + \varepsilon\_it
\end{aligned}
$$

$$(3)$$

Where:

$NDTI\_it$ = Turbidity at site $i$ in year $t$
$Natural_{river}\_it$ = Proportion of natural vegetation in the riparian zone at site $i$, year $t$ (treatment)
$Natural\_watershed\_it$ = Proportion of natural vegetation in the watershed at site $i$, year $t$
$Palm_{river}\_it$ = Proportion of palm plantation in the riparian zone
$Teak_{river}\_it$ = Proportion of teak/gmelina plantation in the riparian zone
$Exposed_{river}\_it$ = Proportion of exposed/urban land in the riparian zone
$GravelRoad_{river}\_it$ = Presence of a gravel road in the riparian zone (binary)
$SoilOrder\_i$ = Riparian zone dominant soil order
$Depth\_i$ = Water depth at the turbidity measurement location
$Discharge\_i$ = River discharge
$UpcurrentNDTI\_it$ = Turbidity at the nearest upcurrent river mouth
$Year\_t$ = Fixed effect for year (1987, 1998, 2019)
$u\_i$ = Random intercept for river site $i$
$\varepsilon\_it$ = Residual error term

These IPW models were run twice: a) with river mouth ID as a random effect and without clustered robust standard errors (RSE), and b) without random effect and with RSE. The models with RSE included the Huber-White correction and site-level clustering. RSE is not frequently used in Ecology research papers, and it accounts for heteroskedasticity, clustered data, correlation between sequential time points, and other correlation structures within datasets[149,151]. RSE make weaker assumptions about the structure of the error and account for serial correlation and arbitrary correlation structures within groups through time, whereas random effects are more efficient but make stronger assumptions about the precise structure of the error. RSE was calculated with the *Sandwich* R package version 3.1.1 clustering on river ID. To assess whether the effects of each statistically significant land use on turbidity varied across rivers with different characteristics, we added interaction terms for river length, slope, and discharge to each original model with a significant treatment effect.

To compare the relative influence of different land uses in different contexts, we evaluated effect sizes from the best-fitting models. These results were later used to infer which spatial scales, land uses, and river types contributed most strongly to downstream turbidity and therefore represent potential priorities for terrestrial management interventions.

## Sensitivity analysis

One of the assumptions of causal inference models is that there are no unobserved confounders. However, in many cases this assumption does not hold. Sensitivity analysis allows for quantitative assessment of how robust the causal estimates are to unobserved confounders[152]. The R package *sensemakr* estimates how omitted confounders would impact the regression result to understand how robust the estimates are to unobserved confounding[94]. *Sensemakr* version 0.1.6 was used for sensitivity analysis for the main IPW models. Models were run as linear models with IPW weights. *Sensemakr* does not allow for mixed-effects models, so the random effect of river mouth ID was not included in the models. *Sensmaker* was run with different variables as the benchmark covariates, the covariates that bound the plausible strength of unobserved confounders. Sensitivity statistics including the partial $R^2$ of treatment with outcome and robustness values were calculated, and the sensitivities were graphed.

To qualitatively evaluate the distribution of the continuous response variables between the treatment groups with and without inverse probability of treatment weighting, we generated side-by-side boxplots[146]. Rivers were considered "treated" if their full riparian zones were more than 50% forested. We used weights from the model of the impact of natural vegetation in full riparian zones to explore the covariate balance with weighting.

**Article**

## Additional causal inference modeling approaches

To assess the robustness of our IPW models and to evaluate how different modeling assumptions affect estimated treatment effects, we compared results across several complementary approaches (see Supplementary Note 2 for a full description of model assumptions). We implemented three additional causal inference approaches designed to reduce bias from unobserved confounding and better handle panel data and nested structure: Two-Way Fixed Effects, Group Mean Covariate, and Group Mean Centering models[59,149]. Two-Way Fixed Effects (TWFE) models were used to take advantage of the panel structure of our data. These models include fixed effects for both river ID and year, allowing us to control for unobserved heterogeneity that is constant over time within river sites (e.g., geomorphology) and time-varying factors shared across rivers (e.g., El Niño/La Niña effects)[149].

Group Mean Covariate Models and Group Mean Centering Models are useful for hierarchical data. In the Group Mean Covariate Model, the group-level mean of a covariate is included as a predictor in the regression analysis to distinguish between the within-group and between-group effects of the covariate. Watersheds were used as groups, as most watersheds had multiple rivers. Group Mean Centering Models subtract the mean value of a variable within a group from each individual's value of that variable. This process creates a new variable that represents the deviation of an individual's value from their group's average. This technique is often used to analyze how individual-level variables relate to an outcome, after accounting for group-level variation[149]. Group Mean Covariate Models and Group Mean Centering Models were clustered based on watersheds as the grouping variable to account for any potential unobserved watershed-level confounders, such as geomorphology or Gulf Coast topography. Thus, these models leverage within-watershed variation across rivers for identification.

Additionally, we ran a naïve linear model using ordinary least squares without accounting for the hierarchical structure of the data or unobserved heterogeneity. We also fit a mixed-effects model that included a random effect for river site to account for correlation among repeated observations within sites.

Clustered RSE with the Huber-White correction and site-level clustering were calculated for all of these models (except the mixed-effects model to maintain the river random effect). For the Two-Way Fixed Effects Models, RSE was calculated using *feols* in the *fixest* R package version 0.12.1. For Group Mean Covariate Models, Group Mean Centering Models, and naïve models, RSE was calculated using the *Sandwich* R package clustering on watersheds.

## Turbidity validation

While previous studies using similar remote sensing products have been published without field validation (e.g. Brown et al., 2017)[5], we validated the relative turbidity values with an *in situ* dataset. *In situ* turbidity was measured by collaborators at 68 locations across the Golfo Dulce in January 2020 and March 2021 (Fig. 1A)[130]. Turbidity was measured at each location at least twice, for a total of 138 observations. All measurements were collected at 0.5 meters depth with a lab-calibrated YSI ProDSS multiparameter probe in Nephelometric Turbidity Units (NTU). Time, data, and weather conditions were measured when measurements were made[130]. For each of the 138 observations, we calculated NDTI at the sampling location in GEE using the complete Landsat 8 imagery collected most proximal to the field NTU measurement. We used the same quality assurance checks and cloud masking in the "Gulf turbidity and depth" section above.

We executed exploratory data analysis to explore the relationship between *in situ* NTU, time and date of *in situ* data collection, and NDTI derived from remotely sensed imagery. Pearson and Spearman correlations between NTU and NDTI were calculated. Linear models and linear mixed effects models were executed, and models were evaluated for whether they abided by ordinary least squares (OLS) assumptions. Root mean squared error (RMSE) and Mean Absolute Error (MAE) were calculated for each model.

## Reporting summary

Further information on research design is available in the Nature Research Reporting Summary linked to this article.

## Data availability

Data used for the analysis and the land use and land cover maps have been deposited in Harvard Dataverse and will be made publicly-available upon publication[153].

## Code availability

Custom code generated for this article was deposited in Harvard Dataverse and may be made available by request to the corresponding author[153].

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

## Acknowledgements

The authors would like to thank the NASA DEVELOP National Program Osa Peninsula Water Resources I-III teams and the Georgia node hosted at the Center for Geospatial Resources, Department of Geography, University of Georgia. We are thankful to Osa Conservation's staff, visitors, and volunteers for their support. We are grateful to Dr. Pablo Gutiérrez Fonseca, Dr. Diane McKnight, Dr. Steve Miller, Blanca Hinojosa, Dr. Joanie Kleypas, Mary Laura Sandoval, Dr. Angels Fernández Mora, and Dr. Rodney Mora Escalante for their guidance. Thank you to the Bobolink Foundation, Gordon and Betty Moore Foundation, Moore Family Foundation, the International Conservation Fund of Canada, Fondation Franklinia, KEEN Effect, and the Troper Wojcicki Foundation for supporting HB, AW, and conservation in Osa. This material is based upon work supported by NASA through contract NNL16AA05C. Any opinions, findings, and conclusions or recommendations expressed in this material are those of the author(s) and do not necessarily reflect the views of the National Aeronautics and Space Administration. This work is supported by the Food and Agricultural Sciences National Needs Graduate and Postgraduate Fellowship Grants Program (NNF), project award no. 2020-38420-30727, from the U.S. Department of Agriculture's National Institute of Food and Agriculture. HB acknowledges the National Science Foundation Graduate Research Fellowship (Grant number: DGE-2146755). LD acknowledges support from the National Science Foundation CAREER #2340606. The graphical abstract was created in BioRender (https://BioRender.com/p9jnczo).

## Author contributions

H.B.: Conceptualization; methodology; software; investigation; validation; formal analysis; writing – original draft; writing – review & editing; visualization; funding acquisition; project administration. L.D.: Methodology; software; formal analysis; writing – review & editing; supervision. H.M.: Methodology; software; investigation; writing – review & editing. J.J.A.B.: Investigation; writing – review & editing. B.B.: Validation; investigation; writing – review & editing. M.G. Bouffard: Methodology; writing – review & editing. MG Burgess: Methodology; supervision; writing – review & editing. J.C.: Investigation; writing – review & editing. S.F.: Methodology; software; validation; writing – review & editing. N.H.: Supervision; data curation; writing – review & editing. A.M.L.: Investigation; Data curation; writing – review & editing. M.M.: Methodology; writing – review & editing; supervision; funding acquisition. E.P.: Methodology; writing – review & editing. R.J.P.S.: Methodology; writing – review & editing. K.J.S.: Methodology; formal analysis; writing – review & editing. L.V.A.: Writing – review & editing; data curation; visualization. A.W.: Writing – review & editing; supervision; funding acquisition. PN: Methodology; writing – review & editing; supervision; funding acquisition.

## Competing interests

The authors declare no competing interests.

## Additional information

[1]Department of Environmental Studies, University of Colorado Boulder, Boulder, CO, USA. [2]Emmett Interdisciplinary Program in Environment and Resources, Doerr School of Sustainability, Stanford University, Stanford, CA, USA. [3]The Natural Capital Project, Stanford University, Stanford, CA, USA. [4]Department of Ecology & Evolutionary Biology, University of Colorado Boulder, Boulder, CO, USA. [5]Energy and Resources Group, University of California Berkeley, Berkeley, CA, USA. [6]Global Policy Laboratory, Stanford Doerr School of Sustainability, Stanford University, Palo Alto, CA, USA. [7]Centro de Investigación en Ciencias del Mar y Limnología (CIMAR),Universidad de Costa Rica, Universidad de Costa Rica, San Pedro, San José, Costa Rica. [8]Escuela de Biología, Universidad de Costa Rica, San José, Costa Rica. [9]Department of Ecology and Evolutionary Biology, School of Biological Sciences, University of Reading, Reading, UK. [10]Spatial Analysis Laboratory, Rubenstein School of Environment & Natural Resources, University of Vermont, Burlington, VT, USA. [11]Department of Economics, University of Wyoming, University Avenue, Laramie, WY, USA. [12]University of Georgia, Athens, GA, USA. [13]Osa Conservation, Puerto Jiménez, Puntarenas, Costa Rica. [14]NASA DEVELOP National Program, Center for Geospatial Research, Department of Geography, University of Georgia, Athens, GA, USA. [15]S&P Global Sustainable1, Commodity Insights, Raleigh, NC, USA. [16]Environmental Studies Program, University of California, Santa Barbara, CA, USA. [17]Department of Geography, University of Colorado Boulder, Boulder, CO, USA. [18]Cooperative Institute for Research in Environmental Science, University of Colorado Boulder, Boulder, CO, USA. [19]Environmental Data Science Innovation & Impact Lab, University of Colorado, Boulde, CO, USA. [20]Department of Biology, Wake Forest University, Winston-Salem, NC, USA. ✉e-mail: hbrum@stanford.edu

