## [Transparent Peer Review file · Communications Sustainability]

Riparian vegetation reduces coastal turbidity

Corresponding Author: Ms Hilary Brumberg

Version 0:

Decision Letter:

Dear Ms Brumberg,

Your manuscript titled "Canopy to coral: Riparian buffers reduce coastal turbidity and protect marine ecosystems" has now been seen by 3 reviewers, whose comments are appended below. You will see that they find your work of some potential interest. However, they have raised quite substantial concerns that must be addressed. In light of these comments, we cannot accept the manuscript for publication, but would be interested in considering a revised version that fully addresses these serious concerns.

We hope you will find the reviewers' comments useful as you decide how to proceed. Additionally, the following editorial thresholds should be addressed in the revised manuscript for further consideration in Communications Earth & Environment:

- * Present compelling new evidence on the role of riparian management over the existing body of literature.
- * If you are not demonstrating the adverse effects on the corals or on any other components of coral ecosystem with experimental evidence, we advise to limit the discussion on coral components.
- * Clearly discuss the methodology and the limitation of the study, and its implications on the results and discussion.

Should additional work allow you to address these criticisms, we would be happy to look at a substantially revised manuscript. If you choose to take up this option, please either highlight all changes in the manuscript text file, or provide a list of the changes to the manuscript with your responses to the reviewers.

When resubmitting, please provide a point-by-point response to the reviewers' comments. Please submit your responses as a separate file, distinct from your cover letter where you can add responses to the Editors' comments that you do not want to be made available to the reviewers. Word files are preferred. We recommend that any figures, tables or graphs that are included in the response to reviewers are also included in the main article or Supplementary Information.

If the revision process takes significantly longer than three months, we will be happy to reconsider your paper at a later date, as long as nothing similar has been accepted for publication at Communications Sustainability or published elsewhere in the meantime.

Please use the following link to submit your revised manuscript, point-by-point response to the reviewers' comments and editorial thresholds, with a list of your changes to the manuscript text (which should be in a separate document to any cover letter), a tracked-changes version of the manuscript (as a PDF file) and any completed checklist:

Link Redacted

**** This url links to your confidential home page and associated information about manuscripts you may have submitted or be**

reviewing for us. If you wish to forward this email to co-authors, please delete the link to your homepage first **

Please do not hesitate to contact us if you have any questions or would like to discuss the required revisions further. Thank you for the opportunity to review your work.

Best regards,

Somaparna Ghosh, PhD
Consulting Editor, Communications Sustainability
Associate Editor,
Communications Earth & Environment

EDITORIAL POLICIES AND FORMAT

If you decide to resubmit your paper, please ensure that your manuscript complies with our editorial policies and complete and upload the checklist below as a Related Manuscript file type with the revised article:

- Behavioural and social science
- Ecological, evolutionary & environmental sciences
- Life sciences

For your information, you can find some guidance regarding format requirements summarized on the following checklist: (<https://www.nature.com/documents/commsj-phys-style-formatting-checklist-article.pdf>) and formatting guide (<https://www.nature.com/documents/commsj-phys-style-formatting-guide-accept.pdf>).

REVIEWER COMMENTS:

Reviewer #1 (Remarks to the Author):

This paper aims to control for confounding land uses and physical characteristics across four causal inference modeling approaches to isolate and quantify the impacts of land uses on coral reefs.

While the paper attempts to account for the role of confounding variables, it makes assumptions that are not entirely clear, and some methodologies require refinement. Ultimately, I find that the objectives of the study are not fully realized.

The justification of the approach is based on the limitations of existing studies; however, it does not clearly demonstrate how this study advances or addresses these limitations. For instance, the authors highlight the need for multi-ecosystem assessments and suggest that multiple habitats or ecosystems would be included, yet only coral reefs are analyzed, despite mentions of seagrasses.

Additionally, the manuscript appears to overlook a substantial body of literature on land-sea interactions, including studies employing catchment modeling to address confounding variables. The methods presented are somewhat simplistic. They note that current efforts are data-intensive and focus on specific watersheds, but it is not entirely convincing that applying this approach to Gulf Dolce—with limited land use and hydrology data—would significantly improve existing methods.

The authors recognize a limitation regarding the spatial scales at which land use affects water quality; however, they neglect to consider more sophisticated oceanographic and dispersal modeling approaches that have been used in prior research.

Furthermore, the manuscript states that a methodology for prioritizing terrestrial areas for downstream conservation was provided, but the specific analytical steps remain unclear. The study would benefit from a comprehensive review of previous work linking land use changes to downstream effects on marine biodiversity.

Finally, the claim that time series data on land use, land cover, and turbidity were used is misleading, as explained in the Methods section.

Reviewer #2 (Remarks to the Author):

The paper evaluates the role of riparian management providing analysis of the value of prioritizing the riparian zone. There is information in this manuscript that will interest scientists in that field. It builds on an existing body of knowledge. The analysis and modelling is detailed and as far as I can judge appropriate - difficult to do more than just skim through in a review.

These are complicated studies always with limitations and you are open about these limitations which I appreciate. I totally agree with your line 401-404. In my part of the world much talk is around the land water interface but actual action is stymied by political boundaries and governance. Good to see you are addressing that. But see specific comments below – I am uncomfortable with extending your analysis to coral with your limited data and the comparisons with MPAs is a bit beyond me. I think the coral component of your argument beyond a discussion point is overreaching a bit - stick with turbidity

Specific suggestions

Line 13 I do not think you can say what is in the graphical abstract a healthy riparian zone does influence water quality but that does not mean coral will be okay “healthier” , “more resilient” maybe??

Lines 56/57 There is a massive amount of work on gully erosion and riparian effects for the Great Barrier Reef – see <https://reefwqconsensus.com.au/question/3-4> and <https://www.reefplan.qld.gov.au/> or any google scholar search see https://scholar.google.com.au/scholar?hl=en&as_sdt=0%2C5&q=Scott+N+Wilkinson1+riprarian&btnG=

I realise these are general statements and much of the Great Barrier Reef is dry tropics and/or intensive sugar cane cropping so different issues but your sweeping statements I think overreach and fail to acknowledge some of the massive coastal “paddock to reef” programs involving hundreds of scientists that exist

Line 82 – I think “aligning” should be aligns

Line 153 - sentence starting Other... is either missing something or just delete “or”

Line 160 – odd wording should it be “than the entire watershed scale”

Line 167/168 advice is unnecessary – words above are sufficient – and everywhere else you advise on other research

Section on Coral - line 209 I have problems with this section – your coral data is very limited and you correctly point out there are many reasons why coral may decline and recover (line231). I am unconvinced that comparing the effect of land use and MPAs is valuable given your data particularly when you compare with Pacific Ocean MPA’s – line 240-241

Lines 246 -251 are okay but the 252-256 is really a poorly supported thought bubble – I suggest the first paragraph (Lines 210-217) is okay – the rest is speculative

Figure 4 – I am not sure I understand what the graph means – maybe a better explanation

Lines 265 to 280 – I am not sure how this fits in with the topic of riparian impacts just off topic ??delete

Line 266 – see links to Great Barrier Reef studies line 56/57

Line 304 consider deleting the last sentence – not relevant to riparian management

Line 398 Mangroves comment is interesting but off topic to riparian influence

Lines 401 to 408 Not sure how this relates to riparian effects

Line 404: “Bridging these divides will contribute to achieving....” Not really “critical”

417-420 - I would delete these lines – not part of your analysis and speculative

Lines 421 – 425 interesting but not on topic – suggest deleting

431 – see previous comments re coral

Line 437 – see previous comments re coral – I think you should limit results to turbidity

Lines 714 to 722 – Given the limited sampling of reefs (line 700) this undermines your conclusions Line (436) and the protection of marine ecosystems as per your title

Reviewer #3 (Remarks to the Author):

I enjoyed reading the paper and think it is a valuable contribution. The paper clearly demonstrates land-based sedimentation impacts coastal corals and that different land-uses deliver different sediment loads. The methods used to demonstrate these relationships are readily available and should encourage practitioners around the globe to apply this kind of analysis.

The findings are not novel, we have know for decades that sediment run off is a problem for a diverse array of coastal ecosystems. We have also known that short flashy rivers are the main source of sediment. I would argue the question is not about the strength of causal inference, but it is about a willingness to act to resolve downstream problems – this issue is not touched on in the manuscript.

I also found that the paper overstates some issues for instance L3 of the abstract: Yet, empirically establishing causal links between land use, freshwater, and marine ecosystems remains challenging. I am not disagreeing with the approach taken in the paper but the paper does not trace sediment from different land uses (this can be done with compound specific stable isotopes), it does not actually demonstrate adverse effects on the corals or any other component of the coastal ecosystem. This I think the value of this contribution is undermined by this type of over statement. This is important because one of the biggest limitations of the approach is that it is focused on averaging over time, and yet many studies have shown that most sediment events are driven by extreme events – this needs to be considered.

Minor points.

I think the approach and description of the Methods is both useful and informative. I did have a problem with the maps I could not see all the features on the map that were identified in the key. Things because clearer as I zoomed into the map – but not when reading it as a paper.

** Visit Nature Portfolio's author and reviewers' website at www.nature.com/authors for information about policies, services and author benefits**

Communications Sustainability is committed to improving transparency in authorship. As part of our efforts in this direction, we are now requesting that all authors identified as 'corresponding author' create and link their Open Researcher and Contributor Identifier (ORCID) with their account on the Manuscript Tracking System prior to acceptance. ORCID helps the scientific community achieve unambiguous attribution of all scholarly contributions. You can create and link your ORCID from the home page of the Manuscript Tracking System by clicking on 'Modify my Springer Nature account' and following the instructions in the link below. Please also inform all co-authors that they can add their ORCID to their accounts and that they must do so prior to acceptance.
<https://www.springernature.com/gp/researchers/orcid/orcid-for-nature-research>

Version 1:

Decision Letter:

Dear Ms Brumberg,

Your manuscript titled "Riparian vegetation reduces coastal turbidity: A causal, empirical, and scalable analysis of land-sea linkages" has now been seen by our reviewers, whose comments appear below. In light of their advice we are delighted to say that we are happy, in principle, to publish a suitably revised version in Communications Sustainability.

We therefore invite you to revise your paper one last time to address the remaining concerns of our reviewers. At the same time we ask that you edit your manuscript to comply with our format requirements and to maximise the accessibility and therefore the impact of your work.

EDITORIAL REQUESTS:

*****Please take care to match our formatting and policy requirements. We will check revised manuscript and return manuscripts that do not comply. Such requests will lead to delays. *****

SUBMISSION INFORMATION:

OPEN ACCESS:

Communications Sustainability is a fully open access journal. Articles are made freely accessible on publication. For further information about article processing charges, open access funding, and advice and support from Nature Portfolio, please visit <https://www.nature.com/commssustain/open-access>

Communications Sustainability is expected to start publishing its first articles in January 2026.

Link Redacted

Best regards,

Somaparna Ghosh, PhD
Consulting Editor, Communications Sustainability
Associate Editor,
Communications Earth & Environment

REVIEWERS' COMMENTS:

Reviewer #1 (Remarks to the Author):

The authors have revised the manuscript and responded to my comments. Unfortunately, my concerns remain and the answers provided were not sufficient. I do not think the limitations explained are valid; the references used/cited still misses assessments of multi-ecosystem vulnerability to catchment uses - combining both oceanographic and hydrological modelling; the prioritization approach does not exist basically.

Reviewer #2 (Remarks to the Author):

General comments

Manuscript is well written. Outlines the value to the marine environment of riparian protection. Is not novel but is a good example and usefully contributes to a global understanding. Most of my concerns have been addressed – I still have some specific comments but they are minor and mostly a matter of opinion

Specific comments

Line 44 Should it be “stabilize”

Line 98 says you quantify causal relationships but on line 101 you say you do not use direct causal modelling – just comes across as confusing – consider deleting last sentence??

Fig 3 – Graphic quality is poor in my copy

Line 240 Comment is speculative unless you have grain size measurements

Figure 4 Am I missing something – needs better explanation – number of observations vs a proportion??

Lines 451-452 Present words are speculative – try “marine ecosystems; mangrove restoration could therefore be especially impactful” -prioritizing a location is beyond you results??

Line 462 Market based solutions are just one of a myriad of ways to sustain and restore riparian zones – your paper does not evaluate the options so I would suggest deleting this one

Line 470 It is a bit unclear what “divides” you are referring to – but in any case it is not “critical” to those global processes and I would advise deleting this sentence

Line 479 – reference is in full

Line 545 What is “unused agriculture” as a land use

Line 578 Reference Brumberg here is in full

Reviewer #3 (Remarks to the Author):

I think the authors have done a good job of improving the paper and addressing a diverse array of comments.

** Visit Nature Portfolio's author and reviewers' website at <http://www.nature.com/authors> for information about policies, services and author benefits**

REVIEWER COMMENTS:

We thank all three reviewers for their thoughtful comments and suggestions. We made substantial revisions to the manuscript based on this feedback and feel that the paper is much stronger as a result. Below, we respond to each comment in turn. Line numbers refer to those in the tracked changes version of the revised manuscript. We included references cited in our responses at the bottom of this document.

Reviewer #1 (Remarks to the Author):

This paper aims to control for confounding land uses and physical characteristics across four causal inference modeling approaches to isolate and quantify the impacts of land uses on coral reefs.

While the paper attempts to account for the role of confounding variables, it makes assumptions that are not entirely clear, and some methodologies require refinement. Ultimately, I find that the objectives of the study are not fully realized.

Thank you for this comment and for highlighting the need to clarify our modeling assumptions and study objectives. We have made several revisions to enhance clarity and refine our description of the methods and assumptions.

First, we now state the study's three objectives explicitly towards the end of the Introduction (lines 103-106), to clarify the scope and contributions of our work. We state: "The objectives of this paper are to: 1) disentangle the relative influence of riparian versus watershed land use on coastal water quality, 2) apply causal inference methods to isolate land use effects while controlling for confounding physical and land use variables, and 3) demonstrate a scalable empirical framework for quantifying land-sea interactions in data-limited regions."

Second, we have added text in the Methods and Discussion to more clearly direct readers to the Supplementary Information (SI sections "IPW model assumptions" and "Sensitivity analysis results"), where all model assumptions, diagnostics, and sensitivity analyses are detailed. These SI sections describe how we addressed standard causal inference assumptions for IPW (no unobserved confounding, positivity, consistency, and no interference), evaluated covariate balance, and quantified robustness to potential omitted confounding variables. Due to wordcount constraints, we summarize these points only briefly in the main text and provide full methodological details in the SI (lines 839-841: "IPW requires the assumptions of no unmeasured confounding, positivity, no interference, consistency, compliance, and correct specification for causal interpretation (see SI section 2 for discussion of these assumptions.>"). We have also included several references, which detail these established methods and their assumptions¹⁻⁵. In

the Discussion (lines 428-433), we now explain that the consistent results across six complementary models (Fig. 6) and the sensitivity analysis results (SI Fig. 7) indicate that our main findings are robust to reasonable levels of unobserved confounding.

We believe these revisions collectively enhance the clarity of the manuscript and show that our objectives are fully addressed through a transparent framework.

The justification of the approach is based on the limitations of existing studies; however, it does not clearly demonstrate how this study advances or addresses these limitations.

In response to this comment, we now more explicitly demonstrate how our study addresses the limitations of previous studies and thus provides a complimentary approach to existing methods. We added a sentence to the Introduction after we detail the limitations of existing studies to specifically highlight how our contribution helps address these limitations: “Our approach provides a complementary, observation-based method that can be applied where detailed process-based catchment or hydrodynamic models are infeasible, where long-term empirical datasets are unavailable, or where confounding limits their ability to isolate the effects of land use change” (lines 101-103).

We also substantially revised the Discussion section on “Implications for future land-sea interaction research” to discuss how our study addresses each of the limitations and to highlight our complementarity with existing approaches (see the main text for the full references cited list):

“This study employs a novel, replicable methodology that integrates freely-accessible satellite imagery, public databases, and causal inference techniques to assess land-sea interactions across large spatial and temporal scales, particularly in remote or data-limited regions. Specifically, this framework advances land-sea research in four main ways. First, whereas most existing studies rely on correlational statistical analyses⁶⁻⁸ that cannot isolate the specific effects of land use from co-varying biophysical conditions, we explicitly account for confounding factors (ex. slope, soil type, and other land uses) using causal inference. Second, although catchment and hydrodynamic models offer valuable mechanistic insight into sediment export and offshore dispersal, they typically require detailed calibration and validation datasets that are unavailable in many tropical systems, and are rarely validated against long-term observed water quality outcomes.⁹⁻¹⁴ Recent work couples terrestrial catchment/erosion models with hydrodynamic or dispersal models (ex. plume transport, nearshore circulation) to more explicitly link land-based sources to downstream marine exposure, but their application remains constrained in data-limited contexts by

parameterization and validation requirements.^{11,15,16} We complement these process-based approaches by providing long-term, observation-based validation of sediment exposure patterns using multi-decadal satellite records. This empirical framework allows us to test, validate, and generalize patterns predicted by process-based modeling studies using real-world observations.

Third, much of the land-sea interaction and ridge-to-reef management literatures focus on a few watersheds or rivers^{7,8,17–20}. Ground-based studies are not scalable, which can limit generalizability, obscure spatial heterogeneity, and reduce relevance for decision-making.^{7,12,21–24} The use of remote sensing allows us to analyze many catchments simultaneously while leveraging temporal replication to separate persistent patterns from river-specific variability. This scalability is especially important in data-limited regions, where publicly available data may be the only feasible evidence base for local decision-making. Fourth, we show that land-sea relationships are spatially scale-dependent, both terrestrially (riparian vs. watershed extents) and marine (distance offshore). Many previous land-sea assessments assume a single relevant spatial scale,^{15,22,25,26} which may obscure effects that emerge only in narrow riparian buffers, as have been well-documented in the freshwater literature. Our results demonstrate that different land uses became detectable at different specific spatial scales, underscoring the importance of explicitly considering spatial heterogeneity to better capture the full extent of ecological linkages. Together, these contributions provide empirical evidence that strengthens and complements process-based approaches in data-limited coastal systems.

Future research could integrate our empirical approach with process-based catchment or hydrodynamic models to improve predictions of sediment fate and ecological impacts. In particular, coupling empirical turbidity estimates with oceanographic dispersal modeling could help resolve fine-scale variation in exposure among coral, seagrass, and mangrove habitats.” (lines 483-526)

For instance, the authors highlight the need for multi-ecosystem assessments and suggest that multiple habitats or ecosystems would be included, yet only coral reefs are analyzed, despite mentions of seagrasses.

We made four major revisions to address your suggestion to better include multiple ecosystems. First, we now more explicitly include mangrove ecosystems by adding a sub-analysis of the impacts of mangrove cover on gulf turbidity (lines 186-190, lines 530-531, lines 595-597, 860-862, and SI Figure 2).

Second, we have now clarified throughout the revised manuscript that our analysis focuses on coastal turbidity, a key pathway linking terrestrial land use to multiple marine ecosystems (not just

coral reefs), established in the existing literature (lines 27-28). We added a Discussion section “Implications for marine ecosystems,” where we now provide a more in-depth review of the implications of changes in turbidity on marine ecosystem functioning and services (lines 438-480).

Third, to better highlight the ecosystems that are likely affected by the land-sea impacts that we analyzed, we demonstrate that sensitive coral reef and seagrass habitats are both located within the nearshore turbidity zone influenced by riverine sediment inputs. We compiled all known maps of coral reefs and seagrasses in the Golfo Dulce from peer-reviewed literature, Costa Rican government reports, and unpublished sources. We revised Figure 1 in the main text to include this map of seagrasses and the full coral reef extent in the region to more clearly illustrate the multiple ecosystems interacting in this system. We also added SI Figure 1, which includes inset maps to clearly illustrate the spatial relationship between land use, riverine turbidity plumes, mangroves, coral reefs, and seagrasses.

Finally, we have revised the paper title to better reflect the scope of the paper, since our findings are relevant to land-sea linkages in general and to avoid overemphasizing coral reefs. The new title highlights our main causal result and our methodological contributions.

Together, we believe that these revisions clarify both the conceptual and methodological scope of the study and how it advances prior land-sea interaction research; thank you for these suggestions that improved the rigor and clarity of the paper.

Additionally, the manuscript appears to overlook a substantial body of literature on land-sea interactions, including studies employing catchment modeling to address confounding variables. The methods presented are somewhat simplistic. They note that current efforts are data-intensive and focus on specific watersheds, but it is not entirely convincing that applying this approach to Gulf Dulce—with limited land use and hydrology data—would significantly improve existing methods.

We appreciate this comment and have revised the Introduction and Discussion to more clearly situate our work within the land-sea interaction literature (see quotes at the bottom of this response for ease of access). We have now added and highlighted key references which identify advances and remaining gaps in the land-sea interaction literature and catchment modeling

Our study takes a complementary, empirical approach to build on the extensive modeling literature you mention. Most prior land-sea studies rely on modeled sediment budgets or process-based simulations that are data-intensive and spatially constrained to well-monitored catchments. Our team includes scientists who have worked with—and even in some cases helped develop—catchment models, and we very much see these as complimentary approaches. Given the data-

hungry nature of these models for parameterization, they cannot be readily applied in our system or would result in great amounts of uncertainty and untested assumptions.

Therefore, we employ an empirical approach, based on multi-decadal remote sensing datasets and causal inference techniques designed to deal with the complexities of observational data^{2,27} to directly quantify the observed effects of land use on turbidity across both watershed and riparian scales.

Notably, our empirical results align with patterns predicted by established modeling studies^{7,11,12,25,28}, reinforcing their conclusions while providing independent, observation-based validation. The revised text emphasizes this complementarity and clarifies that our approach expands the land-sea interaction toolkit by offering a scalable, data-efficient empirical method applicable where traditional models cannot be implemented.

We also more clearly distinguish between land-sea interaction research and ridge-to-reef management frameworks (lines 33-36), because we had previously erroneously used these fields almost interchangeably, and in some case previously we had referred to the ridge-to-reef literature when we meant to refer to the land-sea interaction literature.

- “Nevertheless, most land-sea interaction research evaluates land use only at watershed scales, and therefore seldom distinguishes riparian from broader catchment effects^{15,22,25,26}. Studies connecting river discharge to marine ecosystems typically do not trace these conditions back to specific upstream land use drivers.^{20,29-31} As a result, the link between where sediment originates within a watershed (ex. riparian areas) and how far its effects propagate throughout coastal-marine waters remains weakly resolved in many land-sea assessments, particularly in regions where coupled catchment-marine models cannot be fully parameterized or validated.^{20,29-32}” (lines 63-69).
- “Catchment and hydrodynamic models have advanced the understanding of sediment export from catchments to the ocean, and have in some cases been used to study the impact of drivers (e.g. land use change) on sediment export.⁹⁻¹⁴ Yet, such models often require extensive field, hydrological, and oceanographic data and are difficult to parameterize or empirically validate in data-limited systems.⁹⁻¹⁴ This hinders the ability to clearly isolate the causal effect of specific land use changes or interventions, especially in remote, data-limited regions with sparse monitoring. Moreover, few modelling efforts include long-term empirical validation of upstream drivers and downstream responses, and parameter uncertainty often remains high.⁹⁻¹⁴” (lines 78-85).
- “Our approach provides a complementary, observation-based method that can be applied where detailed process-based catchment or hydrodynamic models are infeasible, where long-term empirical datasets are unavailable, or where confounding limits their ability to isolate the effects of land use change.” (lines 101-103)

- “Second, although catchment and hydrodynamic models offer valuable mechanistic insight into sediment export and offshore dispersal, they typically require detailed calibration and validation datasets that are unavailable in many tropical systems, and are rarely validated against long-term observed water quality outcomes.⁹⁻¹⁴ Recent work couples terrestrial catchment/erosion models with hydrodynamic or dispersal models (ex. plume transport, nearshore circulation) to more explicitly link land-based sources to downstream marine exposure, but their application remains constrained in data-limited contexts by parameterization and validation requirements^{11,15,16}. We complement these process-based approaches by providing long-term, observation-based validation of sediment exposure patterns using multi-decadal satellite records. This empirical framework allows us to test, validate, and generalize patterns predicted by process-based modeling studies using real-world observations.” (lines 490-502)
- “Future research could integrate our empirical approach with process-based catchment or hydrodynamic models to improve predictions of sediment fate and ecological impacts. In particular, coupling empirical turbidity estimates with oceanographic dispersal modeling could help resolve fine-scale variation in exposure among coral, seagrass, and mangrove habitats.” (lines 528-531)

The authors recognize a limitation regarding the spatial scales at which land use affects water quality; however, they neglect to consider more sophisticated oceanographic and dispersal modeling approaches that have been used in prior research.

Thank you for this suggestion. We have now added text in the revised manuscript acknowledging the role of hydrodynamic and dispersal modeling in advancing an understanding of coastal sediment transport. However, we consulted with experts in the region, and very few sophisticated oceanographic and dispersal models currently exist for the Golfo Dulce region. Developing one from scratch is beyond the scope of this study. For context, we note that many prior ridge-to-reef and land-sea studies^{28,33,34} similarly modeled riverine sediment export without resolving oceanic dispersion. Like those studies, our focus is on linking land-use drivers to nearshore water quality, but instead of models, we deploy an analysis of empirical data from remote sensing.

Instead, our analysis focuses on an empirical assessment of observed turbidity gradients extending offshore, and there linking to land use patterns in their upstream watersheds, which allows us to

infer the spatial reach of land-based sediment effects without requiring a process-based model. Our approach complements rather than replaces oceanographic and dispersal modeling by providing observation-based evidence of how far sediment plumes extend from river mouths under real conditions and connecting them directly to land use drivers. This empirical perspective is particularly useful in data-limited tropical systems where complex hydrodynamic models cannot yet be parameterized. We have clarified this rationale in the Introduction and Discussion (see quotes above in response to your previous comment). We also noted that future work integrating our empirical framework with hydrodynamic models could further refine estimates of sediment dispersal (lines 528-531).

Furthermore, the manuscript states that a methodology for prioritizing terrestrial areas for downstream conservation was provided, but the specific analytical steps remain unclear.

We agree that our prioritization framework could be described more explicitly. Our approach identifies priority areas for terrestrial management based on where land use has the largest causal impact on coastal turbidity, which serves as a proxy for downstream benefits for coastal systems. Specifically, our prioritization logic follows three analytical findings: 1) spatial scale (land use impacts were significant only in riparian zones, not watershed scale), 2) land use class (pasture most increased turbidity), and 3) river characteristics (land use impacts were greatest in short, steep, low-discharge rivers. We added text to the Discussion to explain how these analytical findings translate into a prioritization framework for conservation and restoration under limited resources, “Thus, conservation strategies such as riparian restoration will likely yield the largest water quality benefits in shorter, steeper, and lower-discharge rivers, so these rivers may be prioritized in time and resource-scarce contexts. These impacts are likely to be especially pronounced when reforesting pastures, as this land use most increases turbidity, as discussed above.” (lines 387-390, also see lines 565-568).

The study would benefit from a comprehensive review of previous work linking land use changes to downstream effects on marine biodiversity.

We appreciate this suggestion and have strengthened the manuscript accordingly. We now synthesize key studies that link terrestrial land use to marine biodiversity outcomes, and we have now added a Discussion section on “Implications for marine ecosystems” (lines 437-480). We also cite some more relevant literature in the Introduction (lines 27-28, 47-59, 94-98).

Finally, the claim that time series data on land use, land cover, and turbidity were used is misleading, as explained in the Methods section.

We have land use maps from 1987, 1998 and 2019. We calculated NDTI in Google Earth Engine and extracted values from the river mouths at the years, corresponding with the land use maps (1987, 1998, and 2019) to measure the causal effect of land use on NDTI in each time step. We added a sentence to the Methods to clarify this (lines 815-817). We also rephrased “time series” to “multi-decadal” throughout the manuscript in response to your suggestion.

Reviewer #2 (Remarks to the Author):

The paper evaluates the role of riparian management providing analysis of the value of prioritizing the riparian zone. There is information in this manuscript that will interest scientists in that field.

Thank you. We think it will also interest those in terrestrial and marine conservation fields.

It builds on an existing body of knowledge. The analysis and modelling is detailed and as far as I can judge appropriate - difficult to do more than just skim through in a review.

Thank you. The co-authors include researchers with significant experience in the field of causal inference in ecology and remote sensing. To clarify, we have added additional text on the assumption in the Methods section (lines 814-816) and we more clearly point the readers to a more detailed explanation of the methods, assumptions, and sensitivity tests to the SI sections 1 and 2.

These are complicated studies always with limitations and you are open about these limitations which I appreciate. I totally agree with your line 401-404. In my part of the world much talk is around the land water interface but actual action is stymied by political boundaries and governance. Good to see you are addressing that. But see specific comments below – I am uncomfortable with extending your analysis to coral with your limited data and the comparisons with MPAs is a bit beyond me. I think the coral component of your argument beyond a discussion point is overreaching a bit - - stick with turbidity.

We appreciate you pointing out the limitations of the coral component and MPAs. We have now removed the coral component from the main text (we moved it to the SI as an example of how future research could integrate a coral component). We also revised the title to remove the

reference to coral reefs and instead focus on our main causal results and on our methodological contributions. We have also now removed the MPA comparison.

Specific suggestions

Line 13 I do not think you can say what is in the graphical abstract a healthy riparian zone does influence water quality but that does not mean coral will be okay “healthier”, “more resilient” maybe??

Thank you for this suggestion. We have now revised the graphical abstract to say, “more resilient marine ecosystems.” We have also added illustrations of seagrasses and fish that could be found in the coastal waters where riparian land use affects turbidity to avoid over-emphasizing coral reefs since we removed that component of the analysis. We think that this better reflects our study’s implications.

Lines 56/57 There is a massive amount of work on gully erosion and riparian effects for the Great Barrier Reef – see <https://reefwqconsensus.com.au/question/3-4> and <https://www.reefplan.qld.gov.au/> or any google scholar search see https://scholar.google.com.au/scholar?hl=en&as_sdt=0%2C5&q=Scott+N+Wilkinson1+riprian&btnG=

I realise these are general statements and much of the Great Barrier Reef is dry tropics and/or intensive sugar cane cropping so different issues but your sweeping statements I think overreach and fail to acknowledge some of the massive coastal “paddock to reef” programs involving hundreds of scientists that exist

Thank you for calling our attention to this important body of work. We now cite the papers/reports you recommended and others from the Great Barrier Reef^{9,12,16,35–41}. Many of the findings and recommendations from these papers align with our manuscript. Indeed, the similarities between these very different geographies speak to the potential external validity of our findings. Also, the remote sensing causal inference monitoring methodology that we develop in this paper could help address some of the gaps in the literature in the Great Barrier Reef described by Wilkinson et al. in the “2022 Consensus Statement” you pointed us to.¹² For instance, Wilkinson et al. (2022) identifies that many previous studies do not establish causal links, do not quantify impacts, and study either drivers or outcomes but not both. Our study addresses each of these gaps. Overall, these bodies of literature complement each other nicely.

We integrated the information we learned from these reports throughout our manuscript. For example:

- “Most previous research has focused on well-monitored systems such as the Great Barrier Reef and temperate ecosystems...” (lines 47-48)
- “Even in the Great Barrier Reef, one of the world’s most intensively studied land-sea systems, a recent synthesis (Wilkinson et al., 2022) noted that although the key biophysical drivers of sediment and nutrient export are qualitatively understood, few studies have quantified their relative effects, examined long-term changes, or established causal links between land-use change and marine water quality.” (lines 94-98)
- “Our findings align with patterns observed in the well-studied Great Barrier Reef system, where vegetation degradation and loss of riparian cover drive disproportionate sediment and nutrient export, yet extend these insights by providing empirical, multi-decadal evidence in a data-limited tropical system.” (lines 242-244)
- “Likewise, studies on the Great Barrier Reef show that maintaining low nutrient and sediment inputs can substantially improve coral tolerance to heat stress...” (lines 466-468)
- “Lessons from systems where ridge-to-reef management frameworks are already well developed may also be transferable to similar contexts. For example, the Great Barrier Reef benefits from long-standing policy instruments such as the Reef 2050 Plan and the Reef Water Quality Protection Plan, as well as the eReefs interoperable platform that integrates monitoring, modeling, and decision-support tools” (lines 630-633).

Line 82 – I think “aligning” should be aligns

Corrected. Thank you!

Line 153 - sentence starting Other... is either missing something or just delete “or”

We have now revised this sentence to improve clarity.

Line 160 – odd wording should it be “than the entire watershed scale”

We have now revised this paragraph to improve readability.

Line 167/168 advice is unnecessary – words above are sufficient – and everywhere else you advise on other research

We have now removed this sentence, per your suggestion. We also carefully considered and revised other areas where we provide recommendations for future research.

Section on Coral - line 209 I have problems with this section – your coral data is very limited and you correctly point out there are many reasons why coral may decline and recover (line231). I am unconvinced that comparing the effect of land use and MPAs is valuable given your data particularly when you compare with Pacific Ocean MPA's – line 240-241

We appreciate your concerns and agree with this point. As such, we have now removed the MPA comparison entirely. We thoroughly reviewed all previous literature and consulted with the University of Costa Rica and NGOs working in the Golfo Dulce, but no other coral datasets are available from this remote region. Consequently, we have now removed the entire coral sub-analysis (methods, results, and discussion) from the main text, so it no longer contributes to our interpretation. We have put it in the Supplementary Information (SI) as a small case study to help demonstrate how future research might integrate *in situ* coral health or other marine ecosystem data into an analysis. We make it very clear (SI lines 152-155) that this information is intended to be illustrative to build the narrative of the canopy-to-coral relationship and to help inform future work; no statistical analysis was conducted, and we do not endeavor to assess causality. We are also happy to remove it entirely if you or the Editor feel it is more appropriate to do so.

Lines 246 -251 are okay but the 252-256 is really a poorly supported thought bubble – I suggest the first paragraph (Lines 210-217) is okay – the rest is speculative

We have now removed the paragraph of lines 252-256.

Figure 4 – I am not sure I understand what the graph means – maybe a better explanation

We have now revised the panel and axis labels in the figure to improve clarity. We improved the explanation in the caption. Additionally, we have now revised the wording of the corresponding sentences (first paragraph in section “Forest conservation policy aligns with changes in land use and turbidity”, lines 353-356) and we now cite specific panels within Figure 4 to assist the reader's interpretation.

Lines 265 to 280 – I am not sure how this fits in with the topic of riparian impacts just off topic ??delete

We have retained this section because it connects our empirical findings on land use and turbidity trends to Costa Rica's national forest and riparian conservation policies. To make it more clear

how this paragraph fits in the scope and narrative of the study, we have revised the section to explicitly link the policy context to our main results on riparian impacts and downstream turbidity (lines 347-349). This section may be of interest to those in the conservation policy space.

Line 266 – see links to Great Barrier Reef studies line 56/57

We have added references to the Great Barrier Reef studies you mentioned and we have softened the wording in this sentence to avoid overstating the claim.

Line 304 consider deleting the last sentence – not relevant to riparian management
We have deleted it.

Line 398 Mangroves comment is interesting but off topic to riparian influence

The “natural vegetation” class includes mangroves. In response to this comment, as well as Reviewer 1 above, we now include a sub-analysis for each vegetation class included in that aggregated class, including new text on the specific role of mangroves in sediment retention (lines 186-190, lines 530-531, lines 595-597, 860-862, and SI Figure 2). Mangroves had a similar effect size as the aggregated class.

Lines 401 to 408 Not sure how this relates to riparian effects

We have now deleted/tightened parts of these lines to focus on implications directly related to our study.

Line 404: “Bridging these divides will contribute to achieving...” Not really “critical”

Agreed. We have now softened this wording to “may contribute to.”

417-420 - I would delete these lines – not part of your analysis and speculative

We decided to retain this information because it speaks to the conservation implications of our analysis. We have tightened this to align more directly with the findings of our study.

Lines 421 – 425 Interesting but not on topic – suggest deleting

We appreciate your comment. However, we believe this section strengthens the paper's applied relevance of ridge-to-reef conservation. We have revised the paragraph to more clearly connect our empirical findings on riparian restoration and water quality improvements to potential financing mechanisms that could sustain such conservation actions.

431 – see previous comments re coral

As above: we have now removed the coral analysis and all language suggesting that we analyzed the impact of land use on coral reefs. We do discuss (Methods lines 811-813, Discussion lines 438-480) that the areas where land use affects gulf turbidity are likely to overlap with nearshore coral reef and seagrass habitats.

Line 437 – see previous comments re coral – I think you should limit results to turbidity

Agreed. We have removed these sentences.

Lines 714 to 722 – Given the limited sampling of reefs (line 700) this undermines your conclusions Line (436) and the protection of marine ecosystems as per your title

Agreed. We have now removed the coral reef analysis and instead point (at lines 438-455, 463-471) to the large body of empirical evidence that suspended sediments/turbidity could negatively affect coral health and resilience, and we discuss (at lines 479-480) that further research is needed to link remotely sensed turbidity to coral health. We also revised the title to better reflect the scope of our study's main findings and methodological contribution.

Reviewer #3 (Remarks to the Author):

I enjoyed reading the paper and think it is a valuable contribution. The paper clearly demonstrates land-based sedimentation impacts coastal corals and that different land-uses deliver different sediment loads. The methods used to demonstrate these relationships are readily available and should encourage practitioners around the globe to apply this kind of analysis.

Thank you! We appreciate your feedback.

The findings are not novel, we have known for decades that sediment runoff is a problem for a diverse array of coastal ecosystems. We have also known that short flashy rivers are the main source of sediment. I would argue the question is not about the strength of causal inference, but it is about a willingness to act to resolve downstream problems – this issue is not touched on in the manuscript.

We agree that the general links between sediment runoff and coastal ecosystem degradation are well established, albeit to a lesser degree in the tropics where our study site is situated. The novelty of our study lies not in demonstrating that sediment harms coral reefs, but in empirically connecting coastal water quality to the land use of the larger scale landscape. Our novelty lies in 1) creating an empirical, causal, scalable framework for quantifying land-sea linkages in data-limited tropical systems, 2) disentangling riparian and full watershed land use impacts on marine ecosystems. Unlike most prior modeling or site-based studies, our approach integrates multi-decadal satellite data with causal inference methods to isolate the effects of specific land uses while controlling for confounding physical factors and identify where management interventions are likely to be most effective. We have now further clarified this contribution in the Introduction and Discussion.

We also appreciate your thoughtful point about the importance of willingness to act on ridge-to-reef conservation outcomes on the ground. Our aim in this manuscript is to reduce decision uncertainty by clarifying where and how land-based actions most effectively address downstream turbidity, thereby supporting action.

To make the action pathway more explicit in the revised manuscript, we a) identify high-yield terrestrial interventions and where they are most likely to pay off based on our results, b) connect those actions to an existing policy lever in our study system, c) point practitioners to a practical decision aid, and d) identify policy and data tools that have been leveraged in a well-studied system:

- “Our results suggest that in contexts where time and resources for terrestrial management are limited, restoring natural vegetation in riparian zones—particularly those currently dominated by pasture and located along short, steep, low-discharge rivers—will yield the greatest reductions in marine turbidity.” (lines 565-568)
- “Stricter enforcement of the Costa Rican Forest Law protecting riparian buffers, along with more targeted restoration in degraded riparian areas, will likely help buffer sedimentation threats to coastal marine ecosystems.” (lines 576-578)
- “Decision-support tools can help guide when to act on land; for example, the Fredston-Hermann et al. (2016) decision tree helps practitioners, resource managers, fishers, and other resource users determine whether land use should be considered to achieve coastal

marine conservation goals. In our system, that decision tree indicates that land-based drivers are critical, consistent with our results.” (lines 626-629)

- “Lessons from systems where ridge-to-reef management frameworks are already well developed may also be transferable to similar contexts. For example, the Great Barrier Reef benefits from long-standing policy instruments such as the Reef 2050 Plan and the Reef Water Quality Protection Plan, as well as the eReefs interoperable platform that integrates monitoring, modeling, and decision-support tools.” (lines 630-633)

We now also further emphasize policy and funding implications in the Conclusion (lines 646-652) to help translate findings into implementable steps. In this sense, while the manuscript does not measure willingness to act, it is designed to enable action by indicating where riparian restoration is most likely to reduce turbidity, aligning those actions with an enforceable policy instrument, and directing decision-makers to an existing decision tree. Assessing whether scientific evidence increases willingness to act and uptake by managers and resource users is an important social-behavioral question that sits beyond the scope of the present analysis. We view it as a valuable next step.

I also found that the paper overstates some issues for instance L3 of the abstract: Yet, empirically establishing causal links between land use, freshwater, and marine ecosystems remains challenging. I am not disagreeing with the approach taken in the paper but the paper does not trace sediment from different land uses (this can be done with compound specific stable isotopes), it does not actually demonstrate adverse effects on the corals or any other component of the coastal ecosystem. This I think the value of this contribution is undermined by this type of over statement.

We appreciate your comment, and we have revised our paper to more clearly emphasize our specific contributions and avoid overstatements. Our causal analysis was limited to assessing the impacts of land use on coastal turbidity, not directly on coral or seagrass condition. We have now revised the title to remove the reference to corals to avoid implying that we integrated coral reefs in our causal analysis, and instead we focus the title on our main causal result and our methodological contributions. We have removed the small coral reef data analysis from the main manuscript to avoid confusion. We have carefully revised the Abstract, Introduction, Discussion, and Conclusion to clarify this distinction. Specifically, we now emphasize that the study establishes causal relationships between land use and turbidity, while potential implications for marine ecosystems are inferred based on geographic overlap and the well-documented sensitivity of coral reefs and seagrasses to changes in turbidity. These clarifications ensure that the scope of our findings and the strength of evidence are accurately represented.

This is important because one of the biggest limitations of the approach is that it is focused on averaging over time, and yet many studies have shown that most sediment events are driven by extreme events – this needs to be considered.

We agree that extreme rainfall events contribute disproportionately to sediment export and turbidity, and that our averaging approach may underrepresent short-term peaks. Our goal, however, was to isolate the chronic, background influence of land use on coastal turbidity rather than to model episodic flood events, which are driven primarily by climate variability. Following methods used in prior studies (e.g., Brown et al., 2017), we used multi-decadal mean turbidity summaries to capture persistent spatial differences associated with land use while minimizing the influence of transient hydrological extremes. Notably, our models showed stronger and more consistent land use effects during the dry season, when baseflow and turbidity are less dominated by rainfall intensity and more sensitive to landscape characteristics. This suggests that improved riparian management may help reduce sediment export and improve coastal water quality during periods when climatic drivers are less overwhelming. We also note that cloud cover during the rainy season may reduce satellite data availability, reinforcing the focus on long-term average patterns rather than individual events. These clarifications have been added to the Discussion (lines 405-411) and Methods (lines 803-805) sections.

Minor points.

I think the approach and description of the Methods is both useful and informative. I did have a problem with the maps I could not see all the features on the map that were identified in the key. Things became clearer as I zoomed into the map – but not when reading it as a paper.

We are glad to hear you found the Methods both useful and informative. We remade the map Figure 1 to make the features clearer. This included removing non-essential features to make it easier to see key features and adjusting coloring and size of features. Inspired by your comment about zooming into the map, we also added Figure S1, which includes zoomed-in inset maps to help readers visualize the relationships between key coastal features.

References Cited in Responses

1. Austin, P. C. & Stuart, E. A. Moving towards best practice when using inverse probability of treatment weighting (IPTW) using the propensity score to estimate causal treatment effects in observational studies. *Statistics in Medicine* **34**, 3661–3679 (2015).

2. Byrnes, J. E. K. & Dee, L. E. Causal inference with observational data and unobserved confounding variables. *Ecology Letters* **28**, 2024.02.26.582072 (2025).
3. Chesnaye, N. C. *et al.* An introduction to inverse probability of treatment weighting in observational research. *Clinical Kidney Journal* **15**, 14–20 (2022).
4. Kimmel, K., Dee, L. E., Avolio, M. L. & Ferraro, P. J. Causal assumptions and causal inference in ecological experiments. *Trends in Ecology & Evolution* **36**, 1141–1152 (2021).
5. Ramsey, D. S. L., Forsyth, D. M., Wright, E., McKay, M. & Westbrooke, I. Using propensity scores for causal inference in ecology: Options, considerations, and a case study. *Methods in Ecology and Evolution* **10**, 320–331 (2019).
6. Brown, C. J. *et al.* Tracing the influence of land-use change on water quality and coral reefs using a Bayesian model. *Sci Rep* **7**, 4740 (2017).
7. Carlson, R. R., Foo, S. A. & Asner, G. P. Land Use Impacts on Coral Reef Health: A Ridge-to-Reef Perspective. *Front. Mar. Sci.* **6**, (2019).
8. Oleson, K. *et al.* Chapter 11: Linking Landscape and Seascape Conditions: Science, Tools and Management. in 319–364 (2018).
9. Brodie, J. E. *et al.* Terrestrial pollutant runoff to the Great Barrier Reef: An update of issues, priorities and management responses. *Marine Pollution Bulletin* **65**, 81–100 (2012).
10. Oleson, K. L. L. *et al.* Upstream solutions to coral reef conservation: The payoffs of smart and cooperative decision-making. *Journal of Environmental Management* **191**, 8–18 (2017).
11. Delevaux, J. M. S. *et al.* Scenario planning with linked land-sea models inform where forest conservation actions will promote coral reef resilience. *Sci Rep* **8**, 12465 (2018).
12. Wilkinson, S. N., Murray, B. & Prosser, I. *What Are the Primary Biophysical Drivers of Anthropogenic Sediment and Particulate Nutrient Export to the Great Barrier Reef and How Have These Drivers Changed over Time?* <https://reefwqconsensus.com.au/question/3-4/> (2022).
13. Tulloch, V. J. D. *et al.* Improving conservation outcomes for coral reefs affected by future oil palm development in Papua New Guinea. *Biological Conservation* **203**, 43–54 (2016).
14. Ramos-Scharrón, C. E., McLaughlin, P. & Figueroa-Sánchez, Y. Impacts of unpaved roads on runoff and erosion in a dry tropical setting: Isla De Culebra, Puerto Rico. *J Soils Sediments* **24**, 1420–1430 (2024).
15. Rude, J. *et al.* Ridge to reef modelling for use within land–sea planning under data-limited conditions. *Aquatic Conservation: Marine and Freshwater Ecosystems* **26**, 251–264 (2016).
16. Steven, A. D. L. *et al.* eReefs: An operational information system for managing the Great Barrier Reef. *Journal of Operational Oceanography* **12**, S12–S28 (2019).
17. Bartley, R. *et al.* Relating sediment impacts on coral reefs to watershed sources, processes and management: A review. *Science of The Total Environment* **468–469**, 1138–1153 (2014).
18. Bégin, C. *et al.* Increased sediment loads over coral reefs in Saint Lucia in relation to land use change in contributing watersheds. *Ocean & Coastal Management* **95**, 35–45 (2014).
19. Cortés & Risk, M. J. A reef under siltation stress: Cahuita, Costa Rica. *Bulletin of Marine Science* **36**, 339–356 (1985).

20. Fabricius, K. E., Logan, M., Weeks, S. & Brodie, J. The effects of river run-off on water clarity across the central Great Barrier Reef. *Marine Pollution Bulletin* **84**, 191–200 (2014).
21. Couto, T. B. A. & Sethi, S. A. River-to-sea ecosystem management. *Nat Sustain* **7**, 4–6 (2024).
22. Barceló, M., Vargas, C. A. & Gelcich, S. Land–Sea Interactions and Ecosystem Services: Research Gaps and Future Challenges. *Sustainability* **15**, 8068 (2023).
23. Gomiz-Pascual, J. J. *et al.* The fate of Guadalquivir River discharges in the coastal strip of the Gulf of Cádiz. A study based on the linking of watershed catchment and hydrodynamic models. *Science of The Total Environment* **795**, 148740 (2021).
24. Elledge, A. & Thornton, C. Effect of changing land use from virgin brigalow (*Acacia harpophylla*) woodland to a crop or pasture system on sediment, nitrogen and phosphorus in runoff over 25 years in subtropical Australia. *Agriculture, Ecosystems & Environment* **239**, 119–131 (2017).
25. Delevaux, J. M. S. *et al.* Social–ecological benefits of land–sea planning at multiple scales in Mesoamerica. *Nat Sustain* **7**, 545–557 (2024).
26. Maina, J. *et al.* Human deforestation outweighs future climate change impacts of sedimentation on coral reefs. *Nat Commun* **4**, 1986 (2013).
27. Siegel, K. & Dee, L. E. Foundations and Future Directions for Causal Inference in Ecological Research. *Ecology Letters* **28**, e70053 (2025).
28. Suárez-Castro, A. F. *et al.* Global forest restoration opportunities to foster coral reef conservation. *Global Change Biology* **27**, 5238–5252 (2021).
29. MacNeil, M. A. *et al.* Water quality mediates resilience on the Great Barrier Reef. *Nat Ecol Evol* **3**, 620–627 (2019).
30. Warner, J. C., Armstrong, B., He, R. & Zambon, J. B. Development of a Coupled Ocean–Atmosphere–Wave–Sediment Transport (COAWST) Modeling System. *Ocean Modelling* **35**, 230–244 (2010).
31. Bainbridge, Z. *et al.* Fine sediment and particulate organic matter: A review and case study on ridge-to-reef transport, transformations, fates, and impacts on marine ecosystems. *Marine Pollution Bulletin* **135**, 1205–1220 (2018).
32. Ji, C., Zhang, Y., Nejtgaard, J. C. & Ogashawara, I. Assessment of the sediment load in the pearl river estuary based on land use and land cover changes. *CATENA* **250**, 108726 (2025).
33. Brown, C. J. *et al.* Tracing the influence of land-use change on water quality and coral reefs using a Bayesian model. *Sci Rep* **7**, 4740 (2017).
34. Oleson, K. L. L. *et al.* Upstream solutions to coral reef conservation: The payoffs of smart and cooperative decision-making. *Journal of Environmental Management* **191**, 8–18 (2017).
35. De’ath, G. & Fabricius, K. Water quality as a regional driver of coral biodiversity and macroalgae on the Great Barrier Reef. *Ecol Appl* **20**, 840–850 (2010).
36. Devlin, M. J. *et al.* Water Quality and River Plume Monitoring in the Great Barrier Reef: An Overview of Methods Based on Ocean Colour Satellite Data. *Remote Sensing* **7**, 12909–12941 (2015).

37. Fabricius, K. E., Logan, M., Weeks, S. & Brodie, J. The effects of river run-off on water clarity across the central Great Barrier Reef. *Marine Pollution Bulletin* **84**, 191–200 (2014).
38. MacNeil, M. A. *et al.* Water quality mediates resilience on the Great Barrier Reef. *Nat Ecol Evol* **3**, 620–627 (2019).
39. Schroeder, T. *et al.* Inter-annual variability of wet season freshwater plume extent into the Great Barrier Reef lagoon based on satellite coastal ocean colour observations. *Marine Pollution Bulletin* **65**, 210–223 (2012).
40. State of Queensland. *Reef 2050 Water Quality Improvement Plan 2017-2022*. (2018).
41. Wilkinson, S., Jansen, Watts, Chen & Read. *Techniques for Targeting Erosion Control and Riparian Protection in the Goulburn and Broken Catchments, Victoria*. (2005).

REVIEWERS' COMMENTS:

Reviewer #1 (Remarks to the Author):

The authors have revised the manuscript and responded to my comments. Unfortunately, my concerns remain and the answers provided were not sufficient. I do not think the limitations explained are valid; the references used/cited still misses assessments of multi-ecosystem vulnerability to catchment uses - combining both oceanographic and hydrological modelling; the prioritization approach does not exist basically.

Thank you for your feedback. We have now cited additional references on multi-ecosystem vulnerability to catchment uses, including studies that combine oceanographic and hydrological modelling.¹⁻⁵

Additional References cited:

1. Robson, B. J. *et al.* Enhanced assessment of the eReefs biogeochemical model for the Great Barrier Reef using the Concept/State/Process/System model evaluation framework. *Environmental Modelling & Software* **129**, 104707 (2020).
2. Baird, M. E. *et al.* Impact of catchment-derived nutrients and sediments on marine water quality on the Great Barrier Reef: An application of the eReefs marine modelling system. *Marine Pollution Bulletin* **167**, 112297 (2021).
3. Bhattarai, S., Parajuli, P. & Linhoss, A. Integrated Modeling Approach to Assess Freshwater Inflow Impact on Coastal Water Quality. *Water* **16**, 3012 (2024).
4. Macias, D. *et al.* The overlooked impacts of freshwater scarcity on oceans as evidenced by the Mediterranean Sea. *Nat Commun* **16**, 998 (2025).
5. Hunt, B. P. V. *et al.* Advancing an integrated understanding of land–ocean connections in shaping the marine ecosystems of coastal temperate rainforest ecoregions. *Limnology and Oceanography* **69**, 3061–3096 (2024).

Reviewer #2 (Remarks to the Author):

General comments

Manuscript is well written. Outlines the value to the marine environment of riparian protection. Is not novel but is a good example and usefully contributes to a global understanding. Most of my concerns have been addressed – I still have some specific comments but they are minor and mostly a matter of opinion

Thank you very much! We're glad to hear that you feel the study will be useful. We agree with and have integrated all of your suggestions below.

Specific comments

Line 44 Should it be "stabilize"

Corrected, thanks.

Line 98 says you quantify causal relationships but on line 101 you say you do not use direct causal modelling – just comes across as confusing – consider deleting last sentence??

We deleted the last sentence per your suggestion to avoid confusion.

Fig 3 – Graphic quality is poor in my copy

We are submitting a high resolution version with the final manuscript.

Line 240 Comment is speculative unless you have grain size measurements

We revised this sentence to clarify that this interpretation is based on previous studies.

Figure 4 Am I missing something – needs better explanation – number of observations vs a proportion??

We revised the caption to better explain the data displayed in each panel.

Lines 451-452 Present words are speculative – try "marine ecosystems; mangrove restoration could therefore be especially impactful" -prioritizing a location is beyond your results??

We integrated your revision, thanks.

Line 462 Market based solutions are just one of a myriad of ways to sustain and restore riparian zones – your paper does not evaluate the options so I would suggest deleting this one

We deleted this sentence per your suggestion.

Line 470 It is a bit unclear what “divides” you are referring to – but in any case it is not “critical” to those global processes and I would advise deleting this sentence

We deleted this sentence per your suggestion.

Line 479 – reference is in full

We removed the reference and kept it in just superscript notation.

Line 545 What is “unused agriculture” as a land use

“Unused agriculture” is agricultural land that is between planting cycles, such that spectrally it looks like bare ground. We revised this to “unused/fallow agriculture” to clarify.

Line 578 Reference Brumberg here is in full

We removed the reference and kept it in just superscript notation.

Reviewer #3 (Remarks to the Author):

I think the authors have done a good job of improving the paper and addressing a diverse array of comments.

Thank you. We agree that the paper is stronger thanks to all of the reviewers’ comments.